# CausalTime: Realistically Generated Time-series for Benchmarking of Causal Discovery

**Yuxiao Cheng**[1]* **Ziqian Wang**[1]* **Tingxiong Xiao**[1] **Qin Zhong**[3]
**Jinli Suo**[1,2]† **Kunlun He**[3]*
[1]Department of Automation, Tsinghua University
[2]Institute for Brain and Cognitive Science, Tsinghua University (THUIBCS)
[3]Chinese PLA General Hospital
`cyx22,ziqian-w20@mails.tsinghua.edu.cn`

## Abstract

Time-series causal discovery (TSCD) is a fundamental problem of machine learning. However, existing synthetic datasets cannot properly evaluate or predict the algorithms' performance on real data. This study introduces the CausalTime pipeline to generate time-series that highly resemble the real data and with ground truth causal graphs for quantitative performance evaluation. The pipeline starts from real observations in a specific scenario and produces a matching benchmark dataset. Firstly, we harness deep neural networks along with normalizing flow to accurately capture realistic dynamics. Secondly, we extract hypothesized causal graphs by performing importance analysis on the neural network or leveraging prior knowledge. Thirdly, we derive the ground truth causal graphs by splitting the causal model into causal term, residual term, and noise term. Lastly, using the fitted network and the derived causal graph, we generate corresponding versatile time-series proper for algorithm assessment. In the experiments, we validate the fidelity of the generated data through qualitative and quantitative experiments, followed by a benchmarking of existing TSCD algorithms using these generated datasets. CausalTime offers a feasible solution to evaluating TSCD algorithms in real applications and can be generalized to a wide range of fields. For easy use of the proposed approach, we also provide a user-friendly website, hosted on `www.causaltime.cc`.

## 1 Introduction

Inferring causal structures from time-series, i.e., time-series causal discovery (TSCD), is a fundamental problem in machine learning. It goes beyond prediction or forecasting by revealing the complex interactions buried under multi-variate time-series. Recently, many algorithms have been proposed (Löwe et al., 2022; Li et al., 2020; Wu et al., 2022; Brouwer et al., 2021) and achieved satisfactory performance, i.e., the discovered causal graphs are close to the ground-truth counterparts. Under some settings, the causal discovery results are nearly perfect, with AUROC scores approaching 1.

However, the benchmarks for TSCD algorithms do not suffice for the performance evaluation. First of all, for the statistical significance of the quantitative evaluation results, the datasets need to be improved in terms of quality and quantity. Next, the current datasets are limited to several fields and do not cover wide application directions. More importantly, the datasets with ground-truth causal graphs are synthesized and might deviate from the true data-generating process, so the scores may not reflect the performance on real data (Reisach et al., 2021).

Despite the fact that recent works also propose better benchmarks for time-series causal discovery (Lawrence et al., 2021; Runge et al., 2020), as well as static settings (Göbler et al., 2023; Chevalley et al., 2023a;b), current TSCD algorithms often incorporate three types of datasets: *Numerical*

---

*Equal Contribution
†Corresponding author

*datasets*, e.g., VAR (vector auto-regression) and Lorenz-96 (Karimi & Paul, 2010), are simulated using closed-form equations. Although some of these equations (Lorenz-96) are inspired by real application scenarios, e.g., climate dynamics, they are over-simplified and have very limited generalizability to real-world applications (Runge et al., 2020). *Quasi-real datasets* are composed of time-series generated with manually designed dynamics that mimic real counterparts under a certain scenario. For example, DREAM3 (Prill et al., 2010) is a dataset simulated using gene expression and regulation dynamics, and NetSim (Smith et al., 2011) is generated by simulating interactions between human brain regions under observation of fMRI. The problem with this type of dataset is that it only covers a few research areas with underlying mechanisms relatively clearly known. For fields such as healthcare or finance, it is hard or even impossible to generate realistic time-series with manually designed dynamics. *Real datasets* (such as MoCap (Tank et al., 2022), S&P 100 stock returns (Pamfil et al., 2020)) do not have the above-mentioned problem, but the dealbreaker is that the ground truth causal graph is mostly inaccessible, and we have to resort to some ad hoc explanations. As shown in Tab. 1, currently available benchmarking tools cannot support a comprehensive evaluation of the time-series causal discovery algorithm. Therefore, an approach for generating benchmarks that highly mimic the real data in different scenarios and with true causal graphs is highly demanded.

Table 1: Comparison of benchmarks for time-series causal discovery evaluation.

| Datasets | Numerical | Quasi-real | Real | **CausalTime (Ours)** |
|---|---|---|---|---|
| Realistic Data | Low | Moderate | Very High | High |
| With True Causal Graph | ✓ | ✓ | ✗ | ✓ |
| Generalizable to Diverse Fields | ✗ | ✗ | ✓ | ✓ |

In this work, we propose a novel pipeline capable of generating realistic time-series along with a ground truth causal graph and is generalizable to different fields, named **CausalTime**. The process of generating time-series with a given causal graph can be implemented using the autoregression model, however, pursuing a causal graph that matches the target time-series with high accuracy is nontrivial, especially for the data with little prior knowledge about the underlying causal mechanism. To address this issue, we propose to use a deep neural network to fit the observed data with high accuracy, and then retrieve a causal graph from the network or from prior knowledge that holds high data fidelity. Specifically, we first obtain a hypothesized causal graph by performing importance analysis on the neural network or leveraging prior knowledge, and then split the functional causal model into causal term, residual term, and noise term. The split model can naturally generate time-series matching the original data observations well. It is worth noting that the retrieval of the causal graph is not a causal discovery process and does not necessarily uncover the underlying causal relationship, but can produce realistic time-series to serve as the benchmark of causal discovery algorithms. Our benchmark is open-source and user-friendly, we host our website at `www.causaltime.cc`. Specifically, our contributions include:

- We propose CausalTime, a pipeline to generate realistic time-series with ground truth causal graphs, which can be applied to diverse fields and provide new choices for evaluating TSCD algorithms.
- We perform qualitative and quantitative experiments to validate that the generated time-series preserves the characteristics of the original time-series.
- We evaluate several existing TSCD algorithms on the generated datasets, providing some guidelines for algorithm comparison, choice, as well as improvement.

## 2 RELATED WORKS

**Causal Discovery.** Causal Discovery (or Causal Structural Learning), including static settings and dynamic time-series, has been a hot topic in machine learning and made big progress in the past decades. The methods can be roughly categorized into multiple classes. (i) *Constraint-based approaches*, such as PC (Spirtes & Glymour, 1991), FCI (Spirtes et al., 2000), and PCMCI (Runge et al., 2019b; Runge, 2020; Gerhardus & Runge, 2020), build causal graphs by performing conditional independence tests. (ii) *Score-based learning algorithms* which include penalized Neural Ordinary Differential Eqn.s and acyclicity constraint (Bellot et al., 2022) (Pamfil et al., 2020). (iii) Approaches based on *Additive Noise Model (ANM)* that infer causal graph based on additive noise assumption (Shimizu et al., 2006; Hoyer et al., 2008). ANM is extended by Hoyer et al. (2008) to

nonlinear models with almost any nonlinearities. (iv) *Granger-causality-based* approaches. Granger causality was initially introduced by Granger (1969) who proposed to analyze the temporal causal relationships by testing the help of a time-series on predicting another time-series. Recently, Deep Neural Networks (NNs) have been widely applied to nonlinear Granger causality discovery. (Wu et al., 2022; Tank et al., 2022; Khanna & Tan, 2020; Löwe et al., 2022; Cheng et al., 2023b). (v) *Convergent Cross Mapping (CCM)* proposed by Sugihara et al. (2012) that reconstructs nonlinear state space for nonseparable weakly connected dynamic systems. This approach is later extended to situations of synchrony, confounding, or sporadic time-series (Ye et al., 2015; Benkő et al., 2020; Brouwer et al., 2021). The rich literature in this direction requires effective quantitative evaluation and progress in this direction also inspires designing new benchmarking methods. In this paper, we propose to generate benchmark datasets using causal models.

**Benchmarks for Causal Discovery.** Benchmarking is of crucial importance for algorithm design and applications. Researchers have proposed different datasets and evaluation metrics for causal discovery under both static and time-series settings. (i) *Static settings.* Numerical, quai-real, and real datasets are all widely used in static causal discovery. Numerical datasets include datasets simulated using linear, polynomial, or triangular functions (Hoyer et al., 2008; Mooij et al., 2011; Spirtes & Glymour, 1991; Zheng et al., 2018); Quai-real datasets are generated under physical laws (e.g. double pendulum (Brouwer et al., 2021)) or realistic scenarios (e.g. synthetic twin birth datasets (Geffner et al., 2022), alarm message system for patient monitoring (Scutari, 2010; Lippe et al., 2021), neural activity data (Brouwer et al., 2021), and gene expression data (Van den Bulcke et al., 2006)); Real datasets are less frequently used. Examples include "Old Faithful" dataset on volcano eruptions (Hoyer et al., 2008), and expression levels of proteins and phospholipids in human immune system cell (Zheng et al., 2018). Recently, Göbler et al. (2023) proposes a novel pipeline, i.e., causalAssembly, generating realistic and complex assembly lines in a manufacturing scenario. Chevalley et al. (2023a) and Chevalley et al. (2023b) on the other hand, provides CausalBench, a set of benchmarks on real data from large-scale single-cell perturbation. Although causalAssembly and CausalBench are carefully designed, they are restricted in certain research fields where the dynamics can be easily replicated or the ground truth causal relationships can be acquired by performing interventions. (ii) *Time-series settings.* In time-series settings, widely used numerical datasets include VAR and Lorenz-96 (Tank et al., 2022; Cheng et al., 2023b; Khanna & Tan, 2020; Bellot et al., 2022); quasi-real datasets include NetSim (Löwe et al., 2022), Dream-3 / Dream-4 (Tank et al., 2022), and finance dataset simulated using Fama-French Three-Factor Model (Nauta et al., 2019); real datasets include MoCap dataset for human motion data (Tank et al., 2022), S&P 100 stock data (Pamfil et al., 2020), tropical climate data (Runge et al., 2019b), and complex ecosystem data (Sugihara et al., 2012). Other than these datasets, there are several works providing novel benchmarks with ground-truth causal graphs. CauseMe (Runge et al., 2020; 2019a) provides a platform[1] for numerical, quasi-real, as well as real datasets, which are mainly based on TSCD challenges on climate scenarios. However, although the platform is well-designed and user-friendly, it did not alleviate the tradeoff among fidelity, ground truth availability, and domain generalizability. For example, the numerical datasets in CauseMe are still not realistic, and the ground truth causal graphs for real datasets are still based on domain prior knowledge that may not be correct. (Lawrence et al., 2021) focuses on generating time-series datasets that go beyond CauseMe. Their framework allows researchers to generate numerous data with various properties flexibly. The ground-truth graphs for their generated datasets are exact, but the functional dependencies in Lawrence et al. (2021) are still manually designed and may not reflect real dynamics in natural scenarios. As a result, their generated datasets are still categorized into numerical datasets in Tab. 1, although with far higher flexibility.

Recently, neural networks have been extensively studied for their capability of generating time-series Yoon et al. (2019); Jarrett et al. (2021); Pei et al. (2021); Kang et al. (2020); Zhang et al. (2018); Esteban et al. (2017). However, the time-series generated with these methods are improper for benchmarking causal discovery, since causal graphs are not generated alongside the series. Therefore, we propose a pipeline to generate realistic time-series along with the ground truth causal graphs.

---

[1]causeme.net

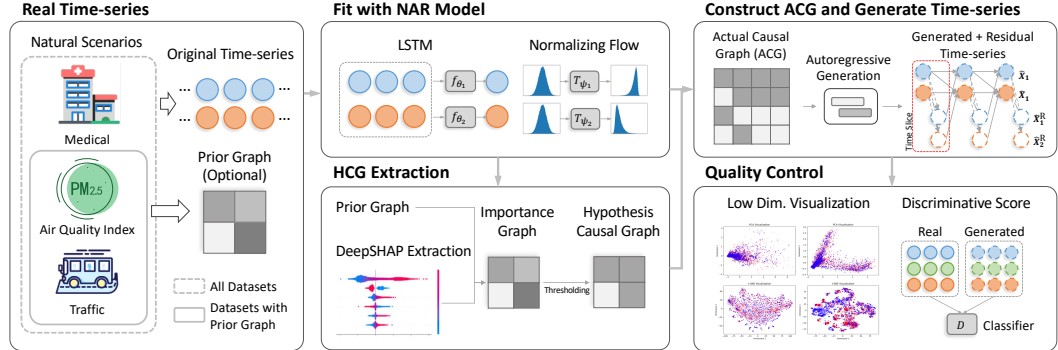

Figure 1: Illustration of the CausalTime pipeline. We fit the observations from real scenarios with NAR model, then split the model and reorganize the components to construct an actual causal graph (ACG) that can generate time-series resembling the input observations. After visual and quantitative quality control, the synthesized time-series and corresponding ACG serve as a benchmark to evaluate the performance of TSCD algorithms in real applications.

## 3 THE PROPOSED TIME-SERIES GENERATION PIPELINE

Aiming at generating time-series that highly resemble observations from real scenarios, and with ground truth causal graphs, we propose a general framework to generate a causal graph built from the real observations and generate counterpart time-series that highly resemble these observations. The whole pipeline consists of several key steps, as illustrated in Fig. 1. Although the real causal graph for the original time-series is unknown, the built causal graph serves as the ground truth lying under the generated version, and they together serve as a benchmark for the TSCD algorithms, shown in Fig. 2. We would like to clarify that, our generation pipeline is based on several assumptions that are common in causal discovery literature: markovian condition, faithfulness, sufficiency, no instantaneous effect, and stationarity. We place the detailed discussion of these assumptions in Supp. Section A.1.1 due to page limits.

### 3.1 CAUSAL MODEL

Causal models in time-series are frequently represented as graphical models (Vowels et al., 2021; Spirtes et al., 2000). However, different from the classic Pearl's causality (Pearl, 2009), spatio-temporal structural dependency must be taken into account for time-series data. We denote a uniformly sampled observation of a dynamic system as $\boldsymbol{X} = \{\boldsymbol{x}_{1:T,i}\}_{i=1}^{N}$, in which $\boldsymbol{x}_t$ is the sample vector at time point $t$ and consists of $N$ variables $\{x_{t,i}\}$, with $t \in \{1, ..., T\}$ and $i \in \{1, ..., N\}$. The structural causal model (SCM) for time-series (Runge et al., 2019b) is $x_{t,i} = f_i\left(\mathcal{P}(x_{t,i}), \eta_{t,i}\right)$, $i = 1, 2, ..., N$, where $f_i$ is any (potentially) nonlinear function, $\eta_{t,i}$ denotes dynamic noise with mutual independence, and $\mathcal{P}(x_t^j)$ denotes the causal parents of $x_t^j$. This model is generalizable to most scenarios, but may bring obstacles for our implementation. In this paper, we consider the nonlinear autoregressive model (NAR), a slightly restricted class of SCM.

**Nonlinear Autoregressive Model.** We adopt the representation in many time-series causal discovery algorithms ((Tank et al., 2022; Löwe et al., 2022; Cheng et al., 2023b)), as well as Lawrence et al. (2021)'s time-series generation pipeline. In a Nonlinear Autoregressive Model (NAR), the noise is assumed to be independent and additive, and each sampled variable $x_{t,i}$ is generated by the following equation:

$$x_{t,i} = f_i\left(\mathcal{P}(x_{t,i})\right) + \eta_{t,i}, \quad i = 1, 2, ..., N. \tag{1}$$

where $\mathcal{P}(\cdot)$ denotes parents in causal graph. We further assume that the maximal time lags for causal effects are limited. Then the model can be denoted as $x_{t,i} = f_i\left(\{x_{\tau,j}\}_{x_{\tau,j} \in \mathcal{P}(x_{t,i})}\right) + \eta_{t,i}$. Here $t - \tau \leq \tau_{\max}, \forall x_{\tau,j} \in \mathcal{P}(x_{t,i})$, and $\tau_{\max}$ denotes the maximal time lag. In causal discovery, time-homogeneity (Gong et al., 2023) is often assumed, i.e., function $f_i$ and causal parents $\mathcal{P}$ is irrelevant to time. By summarizing temporal dependencies, the summary graph for causal models can be denoted with binary matrix $\mathbf{A}$, where its element $a_{ji} = \begin{cases} 1, & \exists \tau, s.t., x_{\tau,j} \in \mathcal{P}(x_{t,i}) \\ 0, & \text{otherwise} \end{cases}$. The dataset pair for causal discovery is $\langle \mathbf{X}, \mathbf{A} \rangle$. TSCD targets to recover matrix $\mathbf{A}$ given time-series

**X**. However, since for most real time-series **X**, causal graph **A** is unknown, benchmarking causal discovery algorithms with real time-series is generally inappropriate.

## 3.2 Time-series Fitting

After collecting real-time-series from diverse fields, we fit the dynamic process of multivariate time-series with a deep neural network and normalizing flow.

**Time-series Fitting with Causally Disentangled Neural Network (CDNN).** To fit the observed time-series with a deep neural network and introduce casual graphs into the network's prediction of output series, Tank et al. (2022); Khanna & Tan (2020); Cheng et al. (2023b) separate the causal effects from the parents to each of individual output series using $N$ separate MLPs / LSTMs, which is referred to as component-wise MLP / LSTM (cMLP / cLSTM). In this paper, we follow (Cheng et al., 2023a)'s definition and refer to the component-wise neural networks as "causally disentangled neural networks" (CDNN), i.e., neural networks taking the form $\mathbf{f}_\Theta(\mathbf{X}, \mathbf{A}) = [f_{\Theta_1}(\mathbf{X} \odot \mathbf{a}_{:,1}), ..., f_{\Theta_n}(\mathbf{X} \odot \mathbf{a}_{:,n})]^T$, where $\mathbf{A} \in \{0,1\}^{n \times n}$ and the operator $\odot$ is defined as $f_{\phi_j}(\mathbf{X} \odot \mathbf{a}_{:,j}) \triangleq f_{\phi_j}(\{\mathbf{x}_1 \cdot a_{1j}, ..., \mathbf{x}_N \cdot a_{Nj}\})$.

So far function $f_{\Theta_j}(\cdot)$ acts as the neural network function used to approximate $f_j(\cdot)$ in Eqn. (1). Since we assume no prior on the underlying causal relationships, to extract the dynamics of the time-series with high accuracy, we fit the generation process with all historical variables (with maximal time lag $\tau_{\max}$, which is discussed in Supp. Section A.1.1) and obtain a fully connected prediction model. Specifically, we assume that

$$x_{t,i} = f_i(\mathbf{x}_{t-\tau:t-1,1}, ..., \mathbf{x}_{t-\tau:t-1,N}) + \eta_{t,i}, \quad i = 1, 2, ..., N. \tag{2}$$

In the following, we omit the time dimension of $\mathbf{x}_{t-\tau:t-1,j}$ and denote it with $\mathbf{x}_j$. Using a CDNN, we can approximate $f_i(\cdot)$ with $f_{\Theta_i}(\cdot)$. (Tank et al., 2022) and (Cheng et al., 2023b) implement CDNN with component-wise MLP / LSTM (cMLP / cLSTM), but the structure is highly redundant because it consists of $N$ distinct neural networks.

**Implementation of CDNN.** The implementation of CDNN can vary. For example, Cheng et al. (2023a) explores enhancing causal discovery with a message-passing-based neural network, which is a special version of CDNN with extensive weight sharing. In this work, we utilize an LSTM-based CDNN with a shared decoder (with implementation details shown in Supp. Section A.2). Moreover, we perform scheduled sampling (Bengio et al., 2015) to alleviate the accumulated error when performing autoregressive generation.

**Noise Distribution Fitting by Normalizing Flow.** After approximating the functional term $f_i(\cdot)$ with $f_{\Theta_i}(\cdot)$, we then approximate noise term $\eta_{t,i}$ with Normalizing Flow (NF) (Kobyzev et al., 2021; Papamakarios et al., 2021). The main process is described as $\hat{\eta}_{t,i} = T_{\psi_i}(u)$, where $u \sim p_u(u)$, in which $T_{\psi_i}(\cdot)$ is an invertible and differentiable transformation implemented with neural network, $p_u(u)$ is the base distribution (normal distribution in our pipeline), and $p_{\hat{\eta}_i}(\hat{\eta}_{t,i}) = p_u(T_{\psi_i}^{-1}(\hat{\eta}_{t,i})) \frac{\partial}{\partial \hat{\eta}_{t,i}} T_{\psi_i}^{-1}(\hat{\eta}_{t,i})$. Then, the optimization problem can be formulated as $\max_{\psi_i} \sum_{t=1}^{N} \log p_u(T_{\psi_i}^{-1}(\eta_{t,i})) + \log \frac{\partial}{\partial \eta_{t,i}} T_{\psi_i}^{-1}(\eta_{t,i})$.

## 3.3 Extraction of Hypothetical Causal Graph

In the fully connected prediction model, all $N$ variables contribute to each prediction, which fits the observations quite well but is over-complicated than the latent causal graph. We now proceed to extract a hypothetical causal graph (HCG) **H** by identifying the most contributing variables in the prediction model $f_{\Theta_i}(\cdot)$. We would like to clarify that, extracting HCG is **not** causal discovery, and it instead targets to identify the contributing causal parents while preserving the fidelity of the fitting model. Two options are included in our pipeline: i) HCG extraction with DeepSHAP; ii) HCG extraction with prior knowledge.

**HCG Extraction with DeepSHAP.** Shapley values (Sundararajan & Najmi, 2020) are frequently used to assign feature importance for regression models. It originates from cooperative game theory (Kuhn & Tucker, 2016), and has recently been developed to interpret deep learning models (DeepSHAP) (Lundberg & Lee, 2017; Chen et al., 2022). For each prediction model $f_{\Theta_i}(\cdot)$, the

calculated importance of each input time-series $\mathbf{x}_{t-\tau:t-1,j}$ by DeepSHAP is $\phi_{ji}$. By assigning importance values from each time-series $j$ to $i$, we get the importance matrix $\Phi$. After we set the sparsity $\sigma$ of a HCG, a threshold $\gamma$ can be calculated with cumulative distribution, i.e., $\gamma = F_\phi^{-1}(\sigma)$, where $F_\phi(\cdot)$ is the cumulative distribution of $\phi_{ji}$ if we assume all $\phi_{ji}$ are i.i.d.

**HCG Extraction with Prior Knowledge.** Time-series in some fields, e.g., weather or traffic, relationships between each variable are highly relevant to geometry distances. For example, air quality or traffic flows in a certain area can largely affect that in a nearby area. As a result, geometry graphs can serve as hypothetical causal graphs (HCGs) in these fields, we show the HCG calculation process in Supp. Section A.2 for this case.

The extracted HCGs is **not the ground truth** causal graph of time-series $\mathbf{X}$, because time-series $\mathbf{X}$ is not generated by the corresponding NAR or SCM model (shown in Fig. 2). A trivial solution would be running auto-regressive generation by setting input of non-causal terms to zero, i.e.,

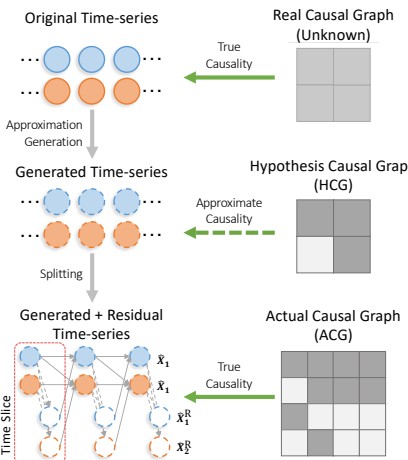

Figure 2: Illustration of how CausalTime gets a $2N \times 2N$ shaped sparse causal graph that exactly synthesizes the desired data.

$$\hat{x}_{t,i} = f_{\Theta_i} \left( \hat{\mathbf{x}}_{t-\tau:t-1,1} \cdot h_{1i}, ..., \hat{\mathbf{x}}_{t-\tau:t-1,N} \cdot h_{Ni} \right) \\ + \hat{\eta}_{t,i}, \quad i = 1, 2, ..., N. \tag{3}$$

where $\{h_{ji}\}$ is the entries in the HCG $\mathbf{H}$. However, the fidelity of the fitting model $f_{\Theta_i}(\cdot)$ in Eq. 3 is hampered by only including a subset of the variables. In the following, we introduce another way to generate time-series with an actual causal graph, and most importantly, without losing fidelity.

## 3.4 Splitting the NAR Model to Acquire Actual Causal Graph and Realistic Time-series

To acquire the Actual Causal Graphs (ACGs) and generate new time-series with high data fidelity, we propose to split the NAR model taking the form of a fully connected prediction model into causal term, residual term, and noise term. In this manner, we do not have to identify exact causal relationships before generating new time-series, and hence avoid the dilemma of "using a TSCD algorithm to build a synthetic dataset to test TSCD algorithms". Specifically,

$$\hat{x}_{t,i} = \underbrace{f_{\Theta_i} \left( \hat{\mathbf{x}}_{t-\tau:t-1,1} \cdot h_{1i}, ..., \hat{\mathbf{x}}_{t-\tau:t-1,N} \cdot h_{Ni} \right)}_{\text{causal term}} + \underbrace{\hat{x}_{t-1,i}^{\text{R}}}_{\text{residual term}} + \underbrace{\hat{\eta}_{t,i}}_{\text{noise term}}, \tag{4}$$

where the residual term $\hat{x}_{t-1,i}^{\text{R}}$ indicates the "causal effect" of non-parent time-series of time-series $i$ in HCG $\mathbf{H}$. In other words, causal terms represent the **"major parts"** of causal effects, and the residual term represents the remaining parts. Mathematically, $\hat{x}_{t-1,i}^{\text{R}}$ is calculated as

$$\hat{x}_{t-1,i}^{\text{R}} = f_{\Theta_i} \left( \hat{\mathbf{x}}_{t-\tau:t-1,1}, ..., \hat{\mathbf{x}}_{t-\tau:t-1,N} \right) - f_{\Theta_i} \left( \hat{\mathbf{x}}_{t-\tau:t-1,1} \cdot h_{1i}, ..., \hat{\mathbf{x}}_{t-\tau:t-1,N} \cdot h_{Ni} \right). \tag{5}$$

and is then stored as an independent time-series. When treated as a generation model, $\mathbf{x}_j \rightarrow \mathbf{x}_i^{\text{R}}$ contains instantaneous effects, however, which does not affect the causal discovery result of $\mathbf{x}_j \rightarrow \mathbf{x}_i$ for most existing TSCD approaches, as we discussed in Supp. Section A.1.2. Indeed, combining Eqn.s 4 and 5, all components of $\mathbf{x}_{t-1}$ are included to predict $\mathbf{x}_t$, i.e., "dense" prediction model. However, the "dense" prediction models in our pipeline do not imply that the "natural" causal graphs are dense. The exact form of the "natural" causal graph are unknown since we do not perform TSCD.

After randomly selecting an initial sequence $\mathbf{X}_{t_0 - \tau_{\max}:t_0-1}$ from the original time-series $\mathbf{X}$, $\mathbf{x}_i$ and $\mathbf{x}_i^{\text{R}}$ is generated via the auto-regressive model, i.e., the prediction results from the previous time step are used for generating the following time step. Our final generated time-series include all $\mathbf{x}_i$ and $\mathbf{x}_i^{\text{R}}$, i.e., a total of $2N$ time-series are generated, and the ACG $\hat{\mathbf{A}}$ is of size $2N \times 2N$:

$$\hat{\mathbf{X}} = \left\{ \hat{\mathbf{X}}_{t_0:t_0+T}; \hat{\mathbf{X}}_{t_0:t_0+T}^{\text{R}} \right\}, \quad \hat{\mathbf{A}} = \begin{bmatrix} \mathbf{H} & \mathbf{J}_N \\ \mathbf{I}_N & \mathbf{0} \end{bmatrix}_{2N \times 2N} \tag{6}$$

where $\mathbf{J}_N$ is an all-one matrix, $\mathbf{I}_N$ is an identity matrix, $T$ is the total length of the generated time-series. Although ground truth causal graphs for the original time-series are unknown, but for the generated version the causal graph is known and unique, shown in Fig. 2.

With the prediction model $f_{\theta_i}$, normalizing flow model $T_{\psi_i}$, and ACG $\mathbf{A}$, we can obtain the final dataset $\left\langle \hat{\mathbf{X}}, \hat{\mathbf{A}} \right\rangle$. $\hat{\mathbf{X}}$ is then fed to TSCD algorithms to recover matrix $\mathbf{A}$ given time-series $\hat{\mathbf{X}}$.

# 4 EXPERIMENTS

In this section, we demonstrate the CausalTime dataset built with the proposed pipeline, visualize and quantify the fidelity of the generated time-series, and then benchmark the performance of existing TSCD algorithms on CausalTime.

## 4.1 STATISTICS OF THE BENCHMARK DATASETS

Theoretically, the proposed pipeline is generalizable to diverse fields. Here we generate 3 types of benchmark time-series from weather, traffic, and healthcare scenarios respectively, as illustrated in Fig. 3. As for the time-series of weather and traffic, relationships between two variables are highly relevant to their geometric distances, i.e., there exists a prior graph, while there is no such prior in the healthcare series. The detailed descriptions are in Supp. Section A.2.

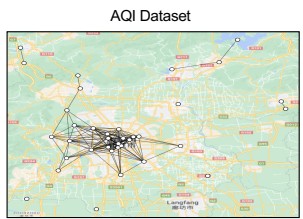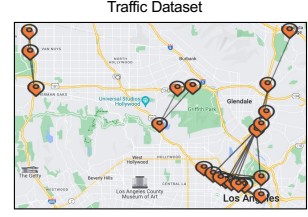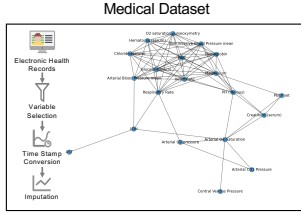

Figure 3: Visualization of three subsets in CausalTime. For AQI and Traffic, we overlay the ground truth causal graphs onto the map.

## 4.2 FIDELITY OF THE GENERATED TIME-SERIES

To qualitatively and quantitatively analyze the fidelity of the generated time-series, we utilize PCA Bryant & Yarnold (1995) and t-SNE van der Maaten & Hinton (2008) dimension reduction visualization, neural-network-based discriminative score, and MMD score of real and synthetic feature vectors to evaluate whether our generated time-series is realistic.

**Visualization via Dimension Reduction.** To judge the fidelity of the generated time-series, we project the time-series features to a two-dimensional space, and assess their similarity by comparing the dimension reduction results. After splitting the original and generated time-series into short sequences (length of 5), we perform dimension reduction via linear (PCA) and nonlinear (t-SNE) approaches on three generated datasets and visualize the difference explicitly, as shown in Fig. 4. One can observe that the distributions of the original and generated series are highly overlapped, and the similarity is especially prominent for AQI and Traffic datasets (i.e., the 1st, 2nd, 4th, and 5th columns). These results visually validate that our generated datasets are indeed realistic across a variety of fields.

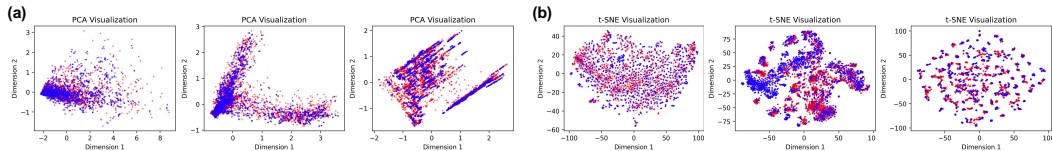

Figure 4: Visualization of the similarity between generated and original time-series in a low dimensional (2D) space, where original and generated series are shown in blue and red, respectively. (a) and (b) plots the two components of PCA and t-SNE on three datasets.

**Discriminative Score / MMD Score.** Other than visualization in low dimensional space, we further assess the generation quality, i.e., evaluate the similarity between the original and

generated time-series, quantitatively using a neural-network-based discriminator and the MMD score. For the neural-network-based discriminator, by labeling the original time-series as positive samples and the generated time-series as negative ones, we train an LSTM classifier and then report the discriminative score in terms of $|\text{AUROC} - 0.5|$ on the test set. MMD is a frequently used metric to evaluate the similarity of two distributions (Gretton et al., 2006). It is estimated with $\widehat{\text{MMD}}_u^2 = \frac{1}{n(n-1)} \sum_{i=1}^{n} \sum_{j \neq i}^{n} K(x_i, x_j) - \frac{2}{mn} \sum_{i=1}^{n} \sum_{j=1}^{m} K(x_i, y_j) + \frac{1}{m(m-1)} \sum_{i=1}^{m} \sum_{j \neq i}^{m} K(y_i, y_j)$, where $K$ is the radial basis function (RBF) kernel. MMD gives another quantitative evaluation of the similarity without the need to train another neural network, as listed in the bottom row of Tab. 2. It is observed that the generated dataset is similar to the original ones, since the discriminative score is very close to zero (i.e., neural networks cannot distinguish generated samples from original samples), and the MMD score is relatively low. Other than discriminative score and MMD, we also utilize cross-correlation scores and perform additive experiments, which is shown in Supp. Section A.3.

Table 2: Quantitative assessment of the similarity between the generated and original time-series in terms of discriminative score and MMD. We show the ablation study in the table as well.

| Datasets | Discriminative Score | | | MMD | | |
| --- | --- | --- | --- | --- | --- | --- |
| | AQI | Traffic | Medical | AQI | Traffic | Medical |
| Additive Gaussian Noise | $0.488 \pm 0.001$ | $0.499 \pm 0.000$ | $0.445 \pm 0.003$ | $0.533 \pm 0.091$ | $0.716 \pm 0.011$ | $0.480 \pm 0.057$ |
| w/o Noise Term | $0.361 \pm 0.008$ | $0.391 \pm 0.006$ | $0.346 \pm 0.001$ | $0.454 \pm 0.025$ | $0.717 \pm 0.007$ | $\mathbf{0.453 \pm 0.029}$ |
| Fit w/o Residual Term | $0.309 \pm 0.010$ | $0.500 \pm 0.000$ | $0.482 \pm 0.005$ | $0.474 \pm 0.033$ | $0.858 \pm 0.020$ | $0.520 \pm 0.023$ |
| Generate w/o Residual Term | $0.361 \pm 0.014$ | $0.371 \pm 0.003$ | $0.348 \pm 0.014$ | $0.431 \pm 0.055$ | $0.657 \pm 0.016$ | $0.489 \pm 0.037$ |
| Full Model | $\mathbf{0.054 \pm 0.025}$ | $\mathbf{0.039 \pm 0.020}$ | $\mathbf{0.017 \pm 0.027}$ | $\mathbf{0.246 \pm 0.029}$ | $\mathbf{0.215 \pm 0.013}$ | $\mathbf{0.461 \pm 0.033}$ |

**Ablation Study.** Using the above two quantitative scores, we also perform ablation studies to justify the effectiveness of our design. In the time-series fitting, we use normalizing flow to fit the noise distributions. To validate its effectiveness, we replace normalizing flow with **(a)** additive Gaussian noise (with parameters estimated from real series) and **(b)** no noise. Further, to validate that our pipeline reserves the real dynamics by splitting the causal model into causal term, residual term, and noise term, we add two alternatives that do not include the residual term when (c) fitting the NAR model or (d) generating new data, besides the above two settings.

The results are shown in Tab. 2, which shows that the full model produces time-series that mimic the original time-series best, in terms of both discriminative score and MMD score. The only exception is the slightly lower MMD under settings "w/o Noise" on medical datasets. It is worth noting that our discrimination scores are close to zero, i.e., neural networks cannot discriminate almost all generated time-series from original versions.

### 4.3 PERFORMANCE OF STATE-OF-THE-ART CAUSAL DISCOVERY ALGORITHMS

To quantify the performances of different causal discovery algorithms, here we calculate their AUROC and AUPRC with respect to the ground truth causal graph. We do not evaluate the accuracy of the discovered causal graph $\tilde{\mathbf{A}}$ with respect to its ground-truth $\hat{\mathbf{A}}$, because there exists instantaneous effects in $\mathbf{x}_j \to \mathbf{x}_i^R$ (see Supp. Section A.1.2). Instead, we ignore the blocks $\mathbf{I}_N, \mathbf{J}_N$ and $\mathbf{0}$ in Eqn. 6), and compare $\tilde{\mathbf{H}}$ with respect to $\mathbf{H}$,

**Baseline TSCD Algorithms.** We benchmarked the performance of 13 recent or representative causal discovery methods on our CausalTime datasets, including: i) Granger-causality-based approach: Granger Causality (GC, (Granger, 1969)), Neural Granger Causality (NGC, (Tank et al., 2022)), economy-SRU (eSRU, (Khanna & Tan, 2020)), Scalable Causal Graph Learning (SCGL, (Xu et al., 2019)), Temporal Causal Discovery Framework (TCDF, (Nauta et al., 2019)), CUTS (Cheng et al., 2023b), CUTS+ (Cheng et al., 2023a) upgrading CUTS to high dimensional time-series. ii) Constraint-based approaches: PCMCI (Runge et al., 2019b), SVAR, NTS-NOTEARS (shown as N.NTS, (Sun et al., 2023)), and Rhino (Gong et al., 2022), iii) CCM-based approaches: Latent Convergent Cross Mapping (LCCM, (Brouwer et al., 2021)), and iv) Other approaches: Neural Graphical Model (NGM, (Bellot et al., 2022)), which employs neural ordinary differential equations to handle irregular time-series data. To ensure fairness, we searched for the best set of hyperpa-

rameters for these baseline algorithms on the validation dataset, and tested performances on testing sets for 5 random seeds per experiment.

Table 3: Performance benchmarking of baseline TSCD algorithms on our CausalTime datasets. We highlight the best and the second best in bold and with underlining, respectively.

| Methods | AUROC | | | AUPRC | | |
|---|---|---|---|---|---|---|
| | AQI | Traffic | Medical | AQI | Traffic | Medical |
| GC | $0.4538_{\pm 0.0377}$ | $0.4191_{\pm 0.0310}$ | $0.5737_{\pm 0.0338}$ | $0.6347_{\pm 0.0158}$ | $0.2789_{\pm 0.0018}$ | $0.4213_{\pm 0.0281}$ |
| SVAR | $0.6225_{\pm 0.0406}$ | $0.6329_{\pm 0.0047}$ | $0.7130_{\pm 0.0188}$ | $0.7903_{\pm 0.0175}$ | $0.5845_{\pm 0.0021}$ | $0.6774_{\pm 0.0358}$ |
| N.NTS | $0.5729_{\pm 0.0229}$ | $\mathbf{0.6329}_{\pm \mathbf{0.0335}}$ | $0.5019_{\pm 0.0682}$ | $0.7100_{\pm 0.0228}$ | $0.5770_{\pm 0.0542}$ | $0.4567_{\pm 0.0162}$ |
| PCMCI | $0.5272_{\pm 0.0744}$ | $0.5422_{\pm 0.0737}$ | $0.6991_{\pm 0.0111}$ | $0.6734_{\pm 0.0372}$ | $0.3474_{\pm 0.0581}$ | $0.5082_{\pm 0.0177}$ |
| Rhino | $0.6700_{\pm 0.0983}$ | $0.6274_{\pm 0.0185}$ | $0.6520_{\pm 0.0212}$ | $0.7593_{\pm 0.0755}$ | $0.3772_{\pm 0.0093}$ | $0.4897_{\pm 0.0321}$ |
| CUTS | $0.6013_{\pm 0.0038}$ | $0.6238_{\pm 0.0179}$ | $0.3739_{\pm 0.0297}$ | $0.5096_{\pm 0.0362}$ | $0.1525_{\pm 0.0226}$ | $0.1537_{\pm 0.0039}$ |
| CUTS+ | $\mathbf{0.8928}_{\pm \mathbf{0.0213}}$ | $0.6175_{\pm 0.0752}$ | $\mathbf{0.8202}_{\pm \mathbf{0.0173}}$ | $\underline{0.7983}_{\pm 0.0875}$ | $\mathbf{0.6367}_{\pm \mathbf{0.1197}}$ | $0.5481_{\pm 0.1349}$ |
| NGC | $0.7172_{\pm 0.0076}$ | $0.6032_{\pm 0.0056}$ | $0.5744_{\pm 0.0096}$ | $0.7177_{\pm 0.0069}$ | $0.3583_{\pm 0.0495}$ | $0.4637_{\pm 0.0121}$ |
| NGM | $0.6728_{\pm 0.0164}$ | $0.4660_{\pm 0.0144}$ | $0.5551_{\pm 0.0154}$ | $0.4786_{\pm 0.0196}$ | $0.2826_{\pm 0.0098}$ | $0.4697_{\pm 0.0166}$ |
| LCCM | $\underline{0.8565}_{\pm 0.0653}$ | $0.5545_{\pm 0.0254}$ | $\underline{0.8013}_{\pm 0.0218}$ | $\mathbf{0.9260}_{\pm \mathbf{0.0246}}$ | $\underline{0.5907}_{\pm 0.0475}$ | $\mathbf{0.7554}_{\pm \mathbf{0.0235}}$ |
| eSRU | $0.8229_{\pm 0.0317}$ | $0.5987_{\pm 0.0192}$ | $0.7559_{\pm 0.0365}$ | $0.7223_{\pm 0.0317}$ | $0.4886_{\pm 0.0338}$ | $\underline{0.7352}_{\pm 0.0600}$ |
| SCGL | $0.4915_{\pm 0.0476}$ | $0.5927_{\pm 0.0553}$ | $0.5019_{\pm 0.0224}$ | $0.3584_{\pm 0.0281}$ | $0.4544_{\pm 0.0315}$ | $\underline{0.4833}_{\pm 0.0185}$ |
| TCDF | $0.4148_{\pm 0.0207}$ | $0.5029_{\pm 0.0041}$ | $0.6329_{\pm 0.0384}$ | $0.6527_{\pm 0.0087}$ | $0.3637_{\pm 0.0048}$ | $0.5544_{\pm 0.0313}$ |

**Results and Analysis.** From the scores in Tab. 3, one can see that among these algorithms, CUTS+ and LCCM perform the best, and most of the TSCD algorithms do not get AUROC > 0.9. The low accuracies tell that current TSCD algorithms still have a long way to go before being put into practice and indicate the necessity of designing more advanced algorithms with high feasibility to real data. Interestingly, a few results demonstrate AUROC < 0.5, which means that we get inverted classifications. Constraint-based approaches such as PCMCI achieve high performance on synthetic datasets (especially linear cases, (Cheng et al., 2023b)) but perform worse on realistic data. This is probably due to the difficulty of conducting conditional independence tests in nonlinear or realistic scenarios. Granger-causality-based approaches such as CUTS+ perform the best because of their neural network-based architecture. However, these methods, such as CUTS, CUTS+, NGC, eSRU, have limitations since they rely on the assumptions of causal sufficiency and no instantaneous effects. CCM-based approach LCCM performs surprisingly well although the performance is not that good in synthetic datasets (reported in Cheng et al. (2023b)). This proves that learning Neural-ODE-based latent processes greatly helps the application of Convergent Cross Mapping. Besides, compared with the reported results in previous work (Cheng et al., 2023b; Tank et al., 2022), one can notice that the scores on our CausalTime dataset are significantly lower than those on synthetic datasets (e.g., VAR and Lorenz-96), on which some TSCD algorithms achieve scores close to 1. This implies that the existing synthetic datasets are insufficient to evaluate the algorithm performance on real data and calls for building new benchmarks to advance the development in this field.

**Additional Information.** We place theoretical analysis and assumptions in Supp. Section A.1, implementation details along with hyperparameters for each of our key steps in Section A.2, additional experimental results (including a comparison of various CDNN implementations, ablation study for scheduled sampling, and experimental results for cross-correlation scores and with missing entries or confounding) in Section A.3, and algorithmic representation in Section 1.

## 5 CONCLUSIONS

We propose CausalTime, a novel pipeline to generate realistic time-series with ground truth causal graphs, which can be used to assess the performance of TSCD algorithms in real scenarios and can also be generalizable to diverse fields. Our CausalTime contributes to the causal-discovery community by enabling upgraded algorithm evaluation under diverse realistic settings, which would advance both the design and applications of TSCD algorithms. Our work can be further developed in multiple aspects. Firstly, replacing NAR with the SCM model can extend CausalTime to a richer set of causal models; secondly, we plan to take into account the multi-scale causal effects that widely exist in realistic time-series. Our future works include i) Creating a more realistic time-series by incorporating prior knowledge of certain dynamic processes, multi-scale associations, or instantaneous effects. ii) Investigation of an augmented TSCD algorithm with reliable results on real time-series data.

## Reproducibility Statement

For the purpose of reproducibility, we include the source code on GitHub (`https://github.com/jarrycyx/UNN`). The generated CausalTime datasets are shared at `www.causaltime.cc`. Moreover, we provide all hyperparameters used for all methods in Appendix Section A.2. All experiments are deployed on a server with Intel Core CPU and NVIDIA RTX3090 GPU.

## Acknowledgments

This work is jointly funded by Ministry of Science and Technology of China (Grant No. 2020AAA0108202), National Natural Science Foundation of China (Grant Nos. 61931012 and 62088102), Beijing Natural Science Foundation (Grant No. Z200021), and Project of Medical Engineering Laboratory of Chinese PLA General Hospital (Grant No. 2022SYSZZKY21, No.2022SYSZZKY11).

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

# A    APPENDIX

## CONTENTS

## A.1    THEORY

### A.1.1    ASSUMPTIONS AND LIMITATIONS

CausalTime is a realistic benchmark to test various TSCD algorithms. As a result, we choose to include the most common assumptions, e.g., Markovian Condition, Causal Faithfulness, No instantaneous effects, Causal Sufficiency, and Causal Stationarity.

**Markovian Condition.** The joint distribution can be factorized into $P(\mathbf{x}) = \prod_i P(x_i | \mathcal{P}(x_i))$, i.e., every variable is independent of all its nondescendants, conditional on its parents.

**Causal Faithfulness.** A causal model that accurately reflects the independence relations present in the data.

**Causal Sufficiency.** (or no latent confounder) All common causes of all variables are observed. This assumption is potentially very strong since we cannot observe "all causes in the world" because it may include infinite variables. However, this assumption is important for a variety of literature.

**No Instantaneous Effect.** (Or Temporal Priority (Assaad et al., 2022)) The cause occurred before its effect in the time-series. This assumption can be satisfied if the sampling frequency is higher than the causal effects. However, this assumption may be strong because the sampling frequency of many real time-series is not high enough. In CausalTime, testing on methods that support instantaneous effects, like Rhino and DYNOTEARS Gong et al. (2022); Pamfil et al. (2020) is still possible by testing only the time-lagging parts. However, it is indeed a limitation that we cannot test the instantaneous part.

**Causal Stationarity.** All the causal relationships remain constant throughout time. With this assumption, full time causal graph can be summarized into a windowed causal graph (Assaad et al., 2022).

In AQI and Traffic datasets, causal relationships are highly relevant to geometry distances, since nearby stations have mutual influences, so the extracted HCG and the subsequent ACG are directly from distance graphs, which align with common sense and are widely used Cini et al. (2022); Wu et al. (2019). In medical datasets, the HCG is extracted with DeepSHAP since it is hard to build reliable graphs from only prior knowledge. Although Shapley values might not exactly match with causality, Shapley-value-based approaches are widely used in the field of medicine and are shown to capture features with actual important relationships Hyland et al. (2020); Thorsen-Meyer et al. (2020). Consequently, the extracted graphs in our three subsets are built on existing extensive studies and are expected to be close to those found in nature.

### A.1.2 Instantaneous Effect of Residual Term

In section 3.4, we propose to split the NAR model into causal term, residual term, and noise term, where the residual term $x_{t-1}^{R}$ is generated with

$$\hat{x}_{t-1,i}^{R} = f_{\Theta_i}\left(\hat{\mathbf{x}}_{t-\tau:t-1,1}, ..., \hat{\mathbf{x}}_{t-\tau:t-1,N}\right) - f_{\Theta_i}\left(\hat{\mathbf{x}}_{t-\tau:t-1,1} \cdot h_{1i}, ..., \hat{\mathbf{x}}_{t-\tau:t-1,N} \cdot h_{Ni}\right) \quad (7)$$

As a result, instantaneous effects exist in this generation equation, i.e., in $\mathbf{x}_j \to \mathbf{x}_i^{R}$ but not in $\mathbf{x}_j \to \mathbf{x}_i$, $\forall i, j$. This is not a problem when tested on TSCD algorithms with compatibility with instantaneous effects. In the following, we discuss the consequences when tested on TSCD algorithms without compatibility to instantaneous effects, e.g., Granger Causality, Convergent Cross Mapping, and PCMCI (Runge et al., 2023). For most cases, this does not affect causal discovery results between all $\mathbf{x}_j$. Though they may draw a wrong conclusion of $\mathbf{x}_j \not\to \mathbf{x}_i^{R}$, this is not the part we compare to ground truth in the evaluation (Section 4.3).

**Granger Causality.** Granger Causality determines causal relationships by testing if a time-series helps the prediction of another time-series. In this case, the causal discovery results of $\mathbf{x}_j \to \mathbf{x}_i$ are not affected since instantaneous effects of $\mathbf{x}_j \to \mathbf{x}_i^{R}$ do not affect whether the time-series $\mathbf{x}_j$ helps to predict the time-series $\mathbf{x}_i$, $\forall i, j$.

**Constraint-based Causal Discovery.** These line of works is based on conditional independence tests, i.e., if $\mathbf{x}_{t_0,j} \to \mathbf{x}_{t,i}$, then $x_{t_0,j} \not\perp x_{t,i} | \mathbf{X}_{t-\tau:t-1} \backslash \{x_{t_0,j}\}$, where $t - \tau \leq t_0 < t$ (Runge et al., 2019b; Runge, 2020). This relationship is unaffected when considering instantaneous effect of $x_{t,j} \to x_{t,i}^{R}$, since all paths from $x_{t_0,j}$ to $x_{t,i}$ through $x_{t,k}^{R}$ are blocked by conditioning on $x_{t,k}^{R}$, $\forall i, j, k$, as is shown in Fig. 5.

**CCM-based Causal Discovery.** CCM detects if time-series $\mathbf{x}$ causes time-series $\mathbf{y}$ by examine whether time indices of nearby points on the $\mathbf{y}$ manifold can be used to identify nearby points on $\mathbf{x}$ (Sugihara et al., 2012; Brouwer et al., 2021). In this case, the examination of whether $\mathbf{x}_j \to \mathbf{x}_i$ is not affected by $\mathbf{x}_k^{R}$, $\forall i, j, k$.

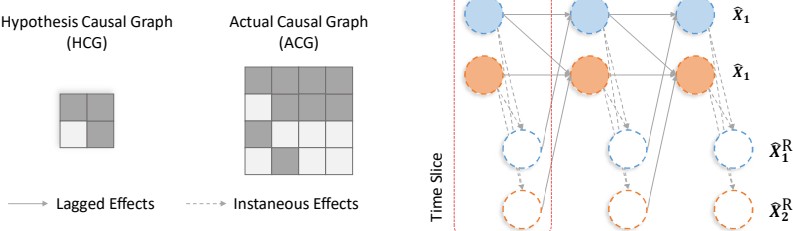

Figure 5: Illustration of instantaneous effects of $\mathbf{x}_j \to \mathbf{x}_i^{R}$. All paths from $x_{t_0,j}$ to $x_{t,i}$ through $x_{t,k}^{R}$ are blocked by conditioning on $x_{t,k}^{R}$, $\forall i, j, k$.

### A.1.3 Causality in The Generated Time-series

The intention of CausalTime is to generate time-series (i) with a unique ground truth causal graph for benchmarking and (ii) highly resembles the distributions of real data for evaluating the algorithm performance on real data. We achieve this in three steps:

1. Extracting HCG in Section 3.3. Here HCG is not the ground truth causal graph since it only reflects the "major part" of the causal relationship.

2. Split the NAR model into causal terms, residual terms, and noise terms. Here causal terms also represent the "major parts" of the causal relationships, and residual term represents the remaining parts.

3. Treat the residual terms $\hat{x}^{R}$ as independent time-series. As a result, there exists $2N$ instead of the original $N$ time-series. And the graph (ACG) becomes $2N \times 2N$, as in Eqn. 6. This process can be visualized in Fig. 2.

In this manner, we can assure that $\hat{x}_{t,i}$ is still generated by exactly the original NAR model since combining Eqn.s 6 and 7 we get the original NAR form. Also, ACG is the ground truth causal graph since we treat the residual terms $\hat{x}^R$ as independent time-series. Hence, ground truth causal graphs for the original time-series are unknown, but for the generated version the causal graph is known and unique, i.e., the ACG. As a result, we avoid facing the dilemma of "using a TSCD algorithm to build a synthetic dataset to test TSCD algorithms". We show this relationship in Fig. 2.

## A.2 IMPLEMENTATION DETAILS

**Original Time-series.** In CausalTime, we generate 3 types of benchmark time-series from weather, traffic, and healthcare scenarios respectively. The original time-series are

1. **Air Quality Index (AQI)** is a subset of several air quality features from 36 monitoring stations spread across Chinese cities[2], with an hourly measurement over one year. We consider the PM2.5 pollution index in the dataset. The total length of the dataset is L = 8760 and the number of nodes is N = 36. We acquire the prior graph by computing the pairwise distances between sensors (Supp. Section A.2).

2. **Traffic** subset is built from the time-series collected by traffic sensors in the San Francisco Bay Area[3]. The total length of the dataset is L = 52116 and we include 20 nodes, i.e., $N = 20$. The prior graph is also calculated with the geographical distance (Supp. Section A.2).

3. **Medical** subset is from MIMIC-4, which is a database that provides critical care data for over 40,000 patients admitted to intensive care units (Johnson et al., 2023). We select 20 most frequently tested vital signs and "chartevents" from 1000 patients, which are then transformed into time-series where each time point represents a 2-hour interval. The missing entries are imputed using the nearest interpolation. For this dataset, a prior graph is unavailable because of the extremely complex dynamics.

**Network Structures and Training.** The implementation of CDNN can vary and cMLP / cLSTM is not the only choice. For example, Cheng et al. (2023a) explores enhancing causal discovery with a message-passing-based neural network, which is a special version of CDNN with extensive weight sharing. However, the fitting accuracy of CDNN is less explored. Sharing partial weights may alleviate the structural redundancy problem. Moreover, the performance with more recent structures is unknown, e.g. Transformer. In the following, this work investigates various implementations of CDNN when they are applied to fit causal models.

Specifically, three kinds of backbones combining three network sharing policies are applied, i.e., MLP, LSTM, Transformer combining no sharing, shared encoder, and shared decoder. For MLP, the encoder and decoder are both MLP. For LSTM, we assign an LSTM encoder and an MLP decoder. For the Transformer, we assign a Transformer encoder and MLP decoder. We show the structure for no sharing, shared encoder, and shared decoder in Fig. 6, with a three-variable example. The test results for prediction using these architectures are shown in Tab. 6.

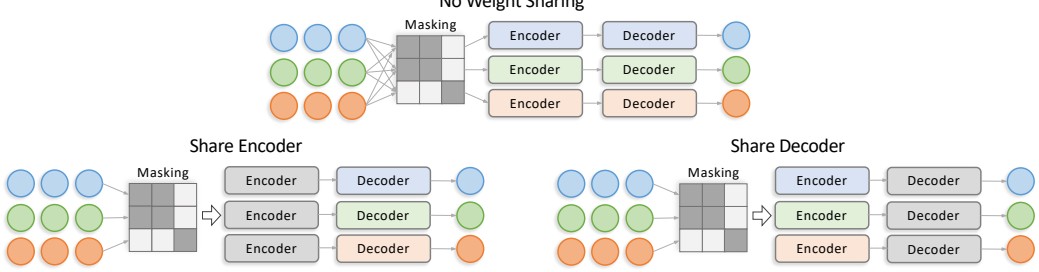

Figure 6: Illustration of network structures of CDNN.

---

[2]https://www.microsoft.com/en-us/research/project/urban-computing/
[3]https://pems.dot.ca.gov/

Table 4: Hyper parameters for time-series fitting.

| Module | Parameter | AQI | Traffic | Medical |
|---|---|---|---|---|
| LSTM Encoder | Layers | 2 | 2 | 2 |
| | Hidden | 128 | 128 | 128 |
| | Heads | 4 | 4 | 4 |
| MLP Encoder | Layers | 3 | 3 | 3 |
| | Hidden | 128 | 128 | 128 |
| Transformer Encoder | Hidden | 128 | 128 | 128 |
| | Heads | 4 | 4 | 4 |
| Decoder | Layers | 3 | 3 | 3 |
| | Hidden | 128 | 128 | 128 |
| Training | Learning Rate | 0.01 | 0.001 | 0.003 |
| | Optimizer | Adam | Adam | Adam |
| | Input Window | 20 | 20 | 20 |
| Normalizing Flow | Layers | 5 | 5 | 5 |
| | Hidden | 128 | 64 | 64 |

**Normalizing Flow.** We implement normalizing flow using its open-source repository[4]. We use normal distributions as base distributions. For transformations, we use simple combinations of linear and nonlinear layers, with parameters shown in Tab. 4.

**DeepSHAP.** We use the official implementation of DeepSHAP[5]. Specifically, we use its "DeepExplainer" module to explain the time-series prediction model which is trained with a fully connected graph. Note that this prediction model is a CDNN, which enables a pairwise explanation of feature $i$'s importance to the prediction of $j$. The explained samples are randomly selected from a train set of real time-series, and the final feature importance graph $\Phi$ is acquired by taking the average values of all samples. To convert to binarized HCG, we select a threshold to get a sparsity of 15% (i.e., 15% of the elements in HCG are labeled as 1).

**Prior Graph Extraction.** For datasets with prior knowledge, e.g., AQI and Traffic, relationships between each variable are highly relevant to geometry distances. Consequently, we extract HCG from the geographic distances between nodes using a thresholded Gaussian kernel, i.e.,

$$w^{ij} = \begin{cases} 1, & \text{dist}(i,j) \leq \sigma \\ 0, & \text{otherwise} \end{cases} \tag{8}$$

we select $\sigma$ based on geographical distances (which is $\approx 40$ km for AQI dataset and for Traffic dataset, we use the "dist_graph" from `https://github.com/liyaguang/DCRNN/tree/master`).

**Dimension Reduction.** For the implementation of t-SNE and PCA, we use scikit-learn[6] package. To solve the dimension reduction in an acceptable time, we split the generated time-series into short sequences (with lengths of 5) and flattened them for the input of dimension reduction.

**Autoregressive Generation.** After fitting the time-series with neural networks and normalizing flow, and acquiring the ground-truth causal graph by splitting the causal model, we generate a new time-series autoregressively. Although we utilize scheduled sampling to avoid the accumulation of the generation error, the total time step must be limited to a relatively small one. Actually, our generated time length is 40 for these three datasets. For each of them, we generate 500 samples, i.e., a total of 20000 time steps for each dataset.

---

[4]https://github.com/VincentStimper/normalizing-flows

[5]https://github.com/shap/shap

[6]https://scikit-learn.org/

Table 5: Hyperparameters settings of the baseline causal discovery and data imputation algorithms.

| Methods | Params. | AQI | Traffic | Medical |
|---|---|---|---|---|
| PCMCI | $\tau_{max}$ | 5 | 5 | 5 |
| | $PC_\alpha$ | 0.05 | 0.05 | 0.05 |
| NGC | Learning rate | 0.05 | 0.05 | 0.05 |
| | $\lambda_{ridge}$ | 0.01 | 0.01 | 0.01 |
| | $\lambda$ | $0.02 \to 0.2$ | $0.02 \to 0.2$ | $0.02 \to 0.2$ |
| eSRU | $\mu_1$ | 0.1 | 0.1 | 0.7 |
| | Learning rate | 0.01 | 0.01 | 0.001 |
| | Batch size | 40 | 40 | 40 |
| | Epochs | 50 | 50 | 50 |
| SCGL | $\tau$ | 10 | 10 | 10 |
| | Batch size | 32 | 32 | 32 |
| | Window | 3 | 3 | 3 |
| LCCM | Epochs | 50 | 50 | 50 |
| | Batch size | 10 | 10 | 10 |
| | Hidden size | 20 | 20 | 20 |
| NGM | Steps | 200 | 200 | 200 |
| | Horizon | 5 | 5 | 5 |
| | GL_reg | 0.05 | 0.05 | 0.05 |
| TCDF | $\tau$ | 10 | 10 | 10 |
| | Epoch num | 1000 | 1000 | 1000 |
| | Learning rate | 0.01 | 0.01 | 0.01 |
| CUTS | Input step | 20 | 20 | 20 |
| | $\lambda$ | 0.1 | 0.1 | 0.1 |
| | $\tau$ | $0.1 \to 1$ | $0.1 \to 1$ | $0.1 \to 1$ |
| CUTS+ | Input step | 1 | 1 | 1 |
| | $\lambda$ | 0.01 | 0.01 | 0.01 |
| | $\tau$ | $0.1 \to 1$ | $0.1 \to 1$ | $0.1 \to 1$ |

**TSCD Algorithm Evaluation.** Since our generated time-series are relatively short and contain several samples, we alter existing approaches by enabling TSCD from multiple observations (i.e. multiple time-series). For neural-network-based or optimization-based approaches such as CUTS, CUTS+, NGC, and TCDF, we alter their dataloader module to prevent cross-sample data fetching. For PCMCI, we use its variant JPCMCI+ which permits the input of multiple time-series. For remaining TSCD algorithms that do not support multiple time-series, we use zero-padding to isolate each sample. We list the original implementations of our included TSCD algorithms in the following:

- *PCMCI.* The code is from `https://github.com/jakobrunge/tigramite`.
- *NGC.* The code is from `https://github.com/iancovert/Neural-GC`. We use the cMLP network because according to the original paper (Tank et al., 2022) cMLP achieves better performance, except for Dream-3 dataset.
- *eSRU.* The code is from `https://github.com/sakhanna/SRU_for_GCI`.
- *SCGL.* The code is downloaded from the link shared in its original paper (Xu et al., 2019).
- *LCCM.* The code is from `https://github.com/edebrouwer/latentCCM`.
- *NGM.* The code is from `https://github.com/alexisbellot/Graphical-modelling-continuous-time`.
- *CUTS / CUTS+.* The code is from `https://github.com/jarrycyx/UNN`.
- *TCDF.* The code is from `https://github.com/M-Nauta/TCDF`.

**Discriminative Network.** To implement the discrimination score in time-series quality control, we train separate neural networks for each dataset to classify the original from the generated time-

Table 6: Comparison of predictive MSE with different implementations of CDNN. The experiments are performed on AQI datasets.

| Backbone | | MLP | LSTM | Transformer |
|---|---|---|---|---|
| Weight Sharing | No Sharing | $0.0511 \pm 0.0009$ | $0.0017 \pm 0.0002$ | $0.0061 \pm 0002$ |
| | Shared Encoder | $0.0248 \pm 0.0033$ | $0.0048 \pm 0.0002$ | $0.0056 \pm 0.0001$ |
| | Shared Decoder | $0.0254 \pm 0.0024$ | $\mathbf{0.0014 \pm 0.0002}$ | $0.0060 \pm 0.0002$ |

series. We use a 2-layered LSTM a with hidden size of 8, the training is performed with a learning rate of 1e-4 and a total of 30 epochs.

## A.3 ADDITIONAL RESULTS

### A.3.1 TIME-SERIES FITTING

In Section 3.2, we show that CDNN can be used to fit causal models with adjacency matrix $\mathbf{A}$. By splitting each network into two parts, i.e., encoder and decoder, 9 combinations (MLP, LSTM, Transformer combining shared encoder, shared decoder, and no weight sharing) are considered in the experiments. By comparing fitting accuracy (or prediction accuracy) on the AQI dataset, we observe that LSTM with a shared decoder performs the best among 9 implementations.

### A.3.2 SCHEDULED SAMPLING

To validate if scheduled sampling is effective in the training process, we perform an ablation study on three datasets with different autoregressive prediction steps. We observe in Tab. 7 that, by incorporating scheduled sampling, the cumulative error decreases, and the decrement is large with higher autoregressive steps. This demonstrates that scheduled sampling does decrease accumulative error for the fitting model, which is beneficial for the following generation process.

Table 7: Comparison of prediction MSE with and without scheduled sampling policy. The comparisons are performed with different autoregressive prediction steps $n$ in terms of MSE.

| Step | AQI | | Traffic | | Medical | |
|---|---|---|---|---|---|---|
| | w | w/o | w | w/o | w | w/o |
| $n = 2$ | 0.0129 | 0.0123 | 0.0124 | 0.0124 | 0.0082 | 0.0082 |
| $n = 5$ | 0.0348 | 0.0329 | 0.0279 | 0.0282 | 0.0136 | 0.0155 |
| $n = 10$ | 0.0351 | 0.0377 | 0.0312 | 0.0354 | 0.0181 | 0.0193 |
| $n = 20$ | 0.0331 | 0.034 | 0.0318 | 0.0331 | 0.0144 | 0.0174 |

### A.3.3 CROSS CORRELATION SCORES FOR TIME-SERIES GENERATION

Despite the discriminative score and MMD score, we further compare the similarity of generated data to original versions in terms of cross-correlation scores. Specifically, we calculate the correlation between real and generated feature vectors, and then report the sum of the absolute differences between them, which is similar to Jarrett et al. (2021)'s calculation process. We show the results along with the ablation study in Tab. 8.

### A.3.4 VARIABILITY OF THE EXTRACTED HCG

For Medical dataset, the HCG is extracted with DeepSHAP and the result may vary with different seed. In Fig. 7 we show the extracted HCG with 5 different seeds, which shows that the variability is relatively small. The average standard deviation is $0.104$. The adjacency matrices of three ACGs of CausalTime are shown in Fig. 8.

Table 8: Quantitative assessment of the similarity between the generated and original time-series in terms of cross-correlation scores. We show the ablation study in the table as well.

| Datasets | Cross Correlation Score | | |
| --- | --- | --- | --- |
| | AQI | Traffic | Medical |
| Additive Gaussian Noise | $43.74 \pm 11.55$ | $198.30 \pm 4.00$ | $18.91 \pm 2.17$ |
| w/o Noise Term | $50.04 \pm 4.79$ | $194.44 \pm 5.02$ | $\mathbf{20.77} \pm \mathbf{4.40}$ |
| Fit w/o Residual Term | $49.04 \pm 7.01$ | $238.38 \pm 4.90$ | $\mathbf{21.94} \pm \mathbf{3.45}$ |
| Generate w/o Residual Term | $40.62 \pm 24.73$ | $164.14 \pm 3.04$ | $23.53 \pm 3.12$ |
| Full Model | $\mathbf{39.75} \pm \mathbf{5.24}$ | $\mathbf{60.37} \pm \mathbf{2.88}$ | $\mathbf{22.37} \pm \mathbf{1.59}$ |

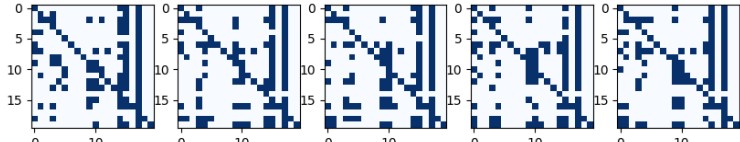

Figure 7: Variability of the extracted HCG.

### A.3.5 TSCD EVALUATION WITH MISSING ENTRIES AND LATENT CONFOUNDING

Missing entries or latent confounding are common in real world TSCD. We simulate these issues on our synthetic data, by dropping some observations at certain missing rates or leaving out certain nodes to create hidden confounding. Specifically, we

- Set random missing rates for the generated dataset, i.e., missing completely at random (MCAR), which is the most common missing scenario (Geffner et al., 2022). Specifically, we use a bi-value observation mask $o_{t,i}$ to label the missing entries. Each data point in the observations are missing with a certain probability $p_i$, here in our experiments it follows Bernoulli distribution $o_{t,i} \sim Ber(1 - p_i)$. Then the time-series with missing data is tested on existing approaches that work with missing entries, i.e., CUTS, CUTS+, LCCM, NGM. We set missing rate to $p = 0.3$ and the result is shown in Tab. 9.

- Leave out certain nodes to create hidden confounding, like Nauta et al. (2019)'s approach. And test existing approaches on the hidden confounding version of CausalTime. Specifically, we deleted node no. 1,17,27,23 for AQI dataset, 7,17 for Traffic dataset, and 6, 9 for Medical dataset. Then we generate new versions of the original ACG. The results are shown in Tab. 10. However, since almost all existing TSCD algorithms struggle to get accurate causal discovery results on CausalTime, it is hard to draw any conclusion by focusing on certain nodes in the confounding version of CausalTime.

Table 9: Performance of TSCD algorithms in the presense of missing data.

| Methods | AUROC | | | AUPRC | | |
| --- | --- | --- | --- | --- | --- | --- |
| | AQI | Traffic | Medical | AQI | Traffic | Medical |
| CUTS | $0.5953 \pm 0.0018$ | $0.5574 \pm 0.0185$ | $0.7864 \pm 0.0400$ | $0.6547 \pm 0.0045$ | $0.7086 \pm 0.0644$ | $0.7672 \pm 0.0074$ |
| CUTS+ | $0.8830 \pm 0.0168$ | $0.5071 \pm 0.0374$ | $0.7296 \pm 0.0155$ | $0.7429 \pm 0.0150$ | $0.5193 \pm 0.0476$ | $0.6774 \pm 0.0134$ |
| NGM | $0.5341 \pm 0.0218$ | $0.6048 \pm 0.0206$ | $0.4328 \pm 0.191$ | $0.7159 \pm 0.0453$ | $0.4596 \pm 0.0342$ | $0.5657 \pm 0.0163$ |
| LCCM | $0.7768 \pm 0.0054$ | $0.5321 \pm 0.0354$ | $0.7877 \pm 0.0661$ | $0.8317 \pm 0.0035$ | $0.5296 \pm 0.0147$ | $0.7699 \pm 0.1175$ |

### A.4 ALGORITHMIC REPRESENTATION FOR CAUSALTIME PIPELINE

We show the detailed algorithmic representation of our proposed data generation pipeline in Algorithm A.4, where we exclude quality control and TSCD evaluation steps.

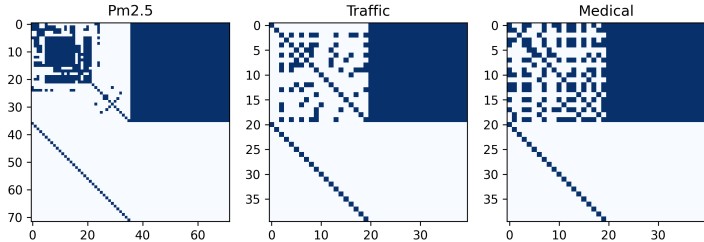

Figure 8: Adjacency matrices of three ACGs in CausalTime.

Table 10: Performance benchmarking of baseline TSCD algorithms on our CausalTime datasets in the presense of hidden confounding.

| Methods | AUROC | | | AUPRC | | |
|---|---|---|---|---|---|---|
| | AQI | Traffic | Medical | AQI | Traffic | Medical |
| GC | $0.4680 \pm 0.0366$ | $0.4414 \pm 0.0375$ | $0.5997 \pm 0.0568$ | $0.6664 \pm 0.0150$ | $0.3160 \pm 0.0054$ | $0.5398 \pm 0.0470$ |
| SVAR | $0.6207 \pm 0.0465$ | $0.63627 \pm 0.0891$ | $0.7478 \pm 0.0282$ | $0.8053 \pm 0.0228$ | $0.6186 \pm 0.0481$ | $0.7733 \pm 0.0267$ |
| N.NTS | $0.4630 \pm 0.0181$ | $0.5238 \pm 0.1268$ | $0.4886 \pm 0.0349$ | $0.6524 \pm 0.0111$ | $0.4165 \pm 0.0416$ | $0.4786 \pm 0.0203$ |
| PCMCI | $0.5865 \pm 0.0673$ | $0.6031 \pm 0.0089$ | $0.7877 \pm 0.0231$ | $0.7385 \pm 0.0284$ | $0.3988 \pm 0.0412$ | $0.6248 \pm 0.0148$ |
| Rhino | $0.7028 \pm 0.0816$ | $0.6327 \pm 0.0233$ | $0.6688 \pm 0.0386$ | $0.7966 \pm 0.0548$ | $0.4286 \pm 0.0218$ | $0.5926 \pm 0.0268$ |
| CUTS | $0.6557 \pm 0.0191$ | $0.7026 \pm 0.0175$ | $0.8184 \pm 0.0069$ | $0.6550 \pm 0.0096$ | $0.7958 \pm 0.0088$ | $0.7886 \pm 0.0071$ |
| CUTS+ | $0.8944 \pm 0.0167$ | $0.6596 \pm 0.0156$ | $0.85507 \pm 0.03069$ | $0.8062 \pm 0.0245$ | $0.7633 \pm 0.0052$ | $0.7585 \pm 0.0036$ |
| NGC | $0.6077 \pm 0.0067$ | $0.6081 \pm 0.0107$ | $0.5637 \pm 0.0023$ | $0.7460 \pm 0.0048$ | $0.4356 \pm 0.0125$ | $0.542 \pm 0.0052$ |
| NGM | $0.5059 \pm 0.0246$ | $0.5540 \pm 0.0278$ | $0.5565 \pm 0.0268$ | $0.7092 \pm 0.0146$ | $0.3886 \pm 0.0185$ | $0.5552 \pm 0.0319$ |
| LCCM | $0.9281 \pm 0.0153$ | $0.6395 \pm 0.0432$ | $0.8185 \pm 0.0329$ | $0.9648 \pm 0.0088$ | $0.6018 \pm 0.0233$ | $0.8465 \pm 0.0255$ |
| eSRU | $0.7473 \pm 0.0438$ | $0.6792 \pm 0.0194$ | $0.8401 \pm 0.0365$ | $0.8634 \pm 0.0271$ | $0.5336 \pm 0.0157$ | $0.8360 \pm 0.0653$ |
| SCGL | $0.5504 \pm 0.0375$ | $0.6434 \pm 0.0497$ | $0.4854 \pm 0.0194$ | $0.3982 \pm 0.0297$ | $0.6009 \pm 0.0173$ | $0.5107 \pm 0.0296$ |
| TCDF | $0.5248 \pm 0.0173$ | $0.5743 \pm 0.0271$ | $0.6995 \pm 0.0243$ | $0.7344 \pm 0.0036$ | $0.4553 \pm 0.0039$ | $0.6375 \pm 0.0145$ |

---

**Algorithm 1** Pipeline for CausalTime Generation (Excluding quality control and TSCD evaluation)

---

**Input:** Time-series dataset $\mathbf{X} = \{\boldsymbol{x}_{1:L,1}, ..., \boldsymbol{x}_{1:L,N}\}$ with length $L$; (Optional) graph $\mathbf{H}_p$ generated with prior knowledge; generation length $\hat{L}$.

**Output:** Paired time-series causal discovery (TSCD) dataset $\left\langle \hat{\mathbf{X}}, \hat{\mathbf{A}} \right\rangle$.

    Initilize parameter set $\{\Theta_i\}_{i=1}^N$, $\{\Psi_i\}_{i=1}^N$
    # Time-series Fitting
    **for** $n_1$ Epochs **do**
        Update $\{\Theta_i\}_{i=1}^N$ with Algorithm 2 (Prediction Model Fitting).
    **end for**
    **for** $n_1$ Epochs **do**
        Update $\{\Psi_i\}_{i=1}^N$ with Algorithm 3 (Noise Distribution Fitting).
    **end for**
    # HCG Extraction
    **if** Exists prior knowledge $\mathbf{H}_p$ **then**
        $\mathbf{H} \leftarrow \mathbf{H}_p$
    **else**
        $\mathbf{H} \leftarrow \{\text{DeepSHAP}\,(f_{\Theta_i}(\cdot))\}_{i=1}^N$
    **end if**
    $\mathbf{H} \leftarrow \mathbb{I}(\mathbf{H} > \gamma)$, where calculation of $\gamma$ is discussed in Section 3.
    # Time-series Generation with ACG
    Generate actual causal graph $\hat{\mathbf{A}}$ with Eqn. 6, select initial sequence $\hat{\mathbf{X}}_{0:\tau-1}$ randomly from original time-series $\mathbf{X}$.
    **for** $t = \tau$ top $\hat{L}$ **do**
        **for** $i = 1$ top $N$ **do**
            Generate $\hat{x}_{t,i}$ and $\hat{x}_{t,i}^{\text{R}}$ with Eqn. 4 and 5, where $\hat{\eta}_{t,i}$ is sampled from $T_{\Psi_i}(u)$.
        **end for**
    **end for**
    **return** Generated TSCD dataset $\left\langle \hat{\mathbf{X}}, \hat{\mathbf{A}} \right\rangle$.

---

**Algorithm 2** Prediction Model Fitting

---

**Input:** Time-series dataset $\{\boldsymbol{x}_{1:L,1}, ..., \boldsymbol{x}_{1:L,N}\}$ with length $L$; parameter set $\{\Theta_i\}_{i=1}^N$.

**Output:** Paired time-series causal discovery (TSCD) dataset $\left\langle \hat{\mathbf{X}}, \hat{\mathbf{A}} \right\rangle$.

    **for** $i = 1$ top $N$ **do**
        Perform prediction with $\hat{x}_{t,i} \leftarrow f_{\Theta_i}(\mathbf{x}_{t-\tau:t-1,1}, ..., \mathbf{x}_{t-\tau:t-1,N}) + \eta_{t,i}$
        Calculate loss function with $\text{MSE}(\hat{X}, X)$
        Update $\{\Theta_i\}_{i=1}^N$ with Adam optimizer
    **end for**
    **return** Discovered causal adjacency matrix $\hat{A}$ where each elements is $\tilde{a}_{i,j}$.

---

**Algorithm 3** Noise Distribution Fitting

---

**Input:** Time-series dataset $\{\boldsymbol{x}_{1:L,1}, ..., \boldsymbol{x}_{1:L,N}\}$ with length $L$; Parameter set $\{\Theta_i\}_{i=1}^N$, $\{\Psi_i\}_{i=1}^N$.

**Output:** Paired time-series causal discovery (TSCD) dataset $\left\langle \hat{\mathbf{X}}, \hat{\mathbf{A}} \right\rangle$.

    **for** $i = 1$ top $N$ **do**
        Perform prediction with $\hat{x}_{t,i} \leftarrow f_{\Theta_i}(\mathbf{x}_{t-\tau:t-1,1}, ..., \mathbf{x}_{t-\tau:t-1,N}) + \eta_{t,i}$.
        Calculate real noise $\eta_{t,i} \leftarrow x_{t,i} - \hat{x}_{t,i}$.
        Calculate likelihood $p_{\eta_i}(\eta_{t,i}) \leftarrow p_u(T_{\psi_i}^{-1}(\eta_{t,i}))\frac{\partial}{\partial \eta_{t,i}}T_{\psi_i}^{-1}(\eta_{t,i})$.
        Update $\{\Psi_i\}_{i=1}^N$ with Adam optimizer.
    **end for**
    **return** Discovered causal adjacency matrix $\hat{A}$ where each elements is $\tilde{a}_{i,j}$.

---

