# OpenReview forum: "CausalTime: Realistically Generated Time-series for Benchmarking of Causal Discovery"
_ICLR.cc/2024/Conference — ICLR 2024 poster_

### Official Review · Reviewer_YwkA · 2023-10-14

**Soundness:** 2 fair
**Presentation:** 2 fair
**Contribution:** 2 fair
**Rating:** 6
**Confidence:** 3

**Summary:**

The paper introduces a pipeline to produce time series that are resembling real world data while being synthetic and analytical enough to serve as benchmarks for TSCD algorithms

**Strengths:**

Based on the evaluation of the method, the produced time series appear to be reasonably realistic.
The method appears to be fairly simple conceptually
There is extensive analysis of the related literature
There is an ablation study - a very welcome addition to the paper.
I believe the community will stand to benefit from using this paper

**Weaknesses:**

The analysis of the method, and the caption of Fig 1 could be improved. Too large emphasis has been given to sounding and appearing mathematical, this makes the  true contribution and impact of the paper, its incorporation in the analysis frameworks of other algorithms, harder to realize as it obfuscates details.
The assumption of stationarity, albeit common, remains very restrictive and should be treated as a limitation. There is a significant amount of real world problems that are non stationary and a method like this would not fare well.

**Questions:**

It is unclear how the method overcomes its main limitation in performance of extracting the causal graph from raw data. It appears that the proposed pipeline uses a  time series causal discovery algorithm, to build a synthetic dataset , to test other  time series causal discovery algorithms, making this an unorthodox loop. How are we guaranteeing the accurate extraction of the underlying DAG to produce the synthetic data.

Real world observational data are rarely well behaved and suffer from , missing entries, confounding, and other biases. A successful  time series causal discovery algorithm needs to be able to disentangle all these from the true causal features. How are the synthetic data taking these biases into account and guarantee not making a too easy task for a TSCD algorithm



### EDIT POST REBUTTAL

I have updated the score from a 5->6

---

> ### Author Response · Authors · 2023-11-15
> **Point-by-point Response to Reviewer YwkA**
>
> Thank you so much for your careful reading and valuable suggestions. They are of great help to improving our manuscript. In the following are the point-to-point responses.
>
> > Q1: The analysis of the method, and the caption of Fig 1 could be improved. Too large emphasis has been given to sounding and appearing mathematical, this makes the true contribution and impact of the paper, its incorporation in the analysis frameworks of other algorithms, harder to realize as it obfuscates details. The assumption of stationarity, albeit common, remains very restrictive and should be treated as a limitation. There is a significant amount of real world problems that are non stationary and a method like this would not fare well.
>
> A1: Many thanks for the advice! We have made the following revisions and updated the manuscript correspondingly:
>
> 1. Update the caption of Fig. 1 to briefly summarize the basic idea and key steps.
> 2. Reduce some mathematical details and update the definition of "CDNN" to help the readers quickly grasp the main idea.
> 3. Add a limitation section in the appendix to show that CausalTime assumes stationarity, causal sufficiency, and no instantaneous effects.
> 4. Add more discussions on the possible impact of the paper in the conclusion section, and more analysis of the existing algorithms in the experiments section.
>
> As for the restriction of stationarity, we would like to clarify that we intend to develop a realistic benchmark to test various TSCD algorithms, some of which support nonstationary time-series while a variety do not (e.g. CUTS, NGC, PCMCI). As a result, we choose to include the most common assumptions, including stationarity.
>
> > Q2: It is unclear how the method overcomes its main limitation in performance of extracting the causal graph from raw data. It appears that the proposed pipeline uses a time series causal discovery algorithm, to build a synthetic dataset , to test other time series causal discovery algorithms, making this an unorthodox loop. How are we guaranteeing the accurate extraction of the underlying DAG to produce the synthetic data.
>
> A2: Thank you for your question and sorry for the confusion. We would like to clarify that CausalTime **does not perform causal discovery when building the synthetic dataset, and instead breaks the iterative loop by fabricating (NOT discovering) a causal graph being able to generate data resembling the real time-series.** Although looks like a causal discovery, the extraction of HCG instead targets to identify the major parts of causal effects and further decompose the NAR model into the main causal term, residual term, and noise. The splitting facilitates getting a $2N\times 2N$ shaped sparse causal graph that exactly synthesizes the desired data. We have added illustrative Fig. 2 to clarify the misunderstanding.
>
> > Q3: Real world observational data are rarely well behaved and suffer from missing entries, confounding, and other biases. A successful time series causal discovery algorithm needs to be able to disentangle all these from the true causal features. How are the synthetic data taking these biases into account and guarantee not making a too easy task for a TSCD algorithm
>
> A3: Helpful comments! Thank you. Indeed, missing entries and confounding are serious problems in causal discovery. **We can also simulate these issues on our synthetic data, by dropping some observations at certain missing rates or leaving out certain nodes to create hidden confounding** (like in [1]). To support this point, we have added experiments to test existing TSCD approaches in the presence of missing entries and unobserved confounding, as shown in Supp. Section A.3.5.
>
> [1] M. Nauta, D. Bucur, and C. Seifert, “Causal discovery with attention-based convolutional neural networks,” Machine Learning and Knowledge Extraction.
>
> Thanks again for your time and we would be happy to answer any additional queries you may have!

---

> > ### Comment · Reviewer_YwkA · 2023-11-17
> > **Acknowledgement of rebuttal**
> >
> > I have read the authors rebuttal , the responses to the questions posed and points raised are satisfactory. Im updating my score in line with this

---

> > > ### Author Response · Authors · 2023-11-17
> > > **Thank you!**
> > >
> > > Thanks for your appreciation and quick reply! This means a lot to us.

---

### Official Review · Reviewer_3QnF · 2023-11-01

**Soundness:** 3 good
**Presentation:** 3 good
**Contribution:** 3 good
**Rating:** 8
**Confidence:** 4

**Summary:**

This paper proposes a methodology for producing realistic time series data with ground truth, which can be used to evaluate algorithms for causal structure learning in time series.  For most domains, it's impossible to have realistic data with a known ground truth causal structure, since we don't know the ground truth dynamics of most domains, so most data for evaluation of causal modeling algorithms is at least semi-synthetic.  The authors' approach builds on this tradition by learning a model of realistic data, treating that model as the ground truth, and using it to generate a new data set that can be used for evaluation.  The authors detail the methodology for their approach and perform a series of experiments, both to assess how realistic the generated data looks and to compare the performance of multiple time series causal discovery algorithms.

**Strengths:**

While the overall idea of this paper (fitting a model on realistic data, and then using it to generate data that has a known ground truth) isn't novel, I haven't seen it applied to time series data, and I think the authors' treatment of it is well-described and motivated.  The authors describe their methodology well and provide a reasonable technical foundation.

The experiments cover a useful breadth.  The comparison of the data distribution between the original and generated time series was interesting, and I appreciated how the ablation study, highlighting the importance of each piece of the equation.  For the comparison of causal discovery algorithms, I thought the authors picked a reasonable set of algorithms to compare, providing a nice demonstration of the application of CausalTime.

**Weaknesses:**

My biggest confusion/concern about this work is the inclusion of the residual term.  Clearly, from Table 2, the residual term plays a massive role in producing data that looks like the original data.  However, it seems as though the presence of the residual term simply means that every time series depends on every other time series.  While it makes sense that including more variables would allow for more accurate model fitting, it's not clear to me that the resulting data actually reflects the causal dynamics of the supposed ground truth graph.  Looking at Equations 6 and 7, it doesn't look like the residual term is down-weighted or anything to reduce its contribution.  Substituting Equation 7 into Equation 6, it actually looks as though the primary f_{theta_i} terms cancel, leaving us with the first term in Equation 7 plus the noise term, which essentially means that we're just ignoring which variables are actually parents of x_i and just including everything.  Am I misunderstanding or misinterpreting something?  It looks to me like a ground truth graph is generated, and realistic-looking data is generated, but the realistic-looking data doesn't actually come from that graph.  This is the main reason my score is a 6 rather than an 8, and if this is cleared up satisfactorally, I'd be happy to raise my score.

A bit more analysis for the final evaluation would be helpful.  The authors point out that, when evaluated on synthetic data in prior work, the scores are higher across the board.  While that's interesting, I'm much more interested in if the conclusions we would draw as a result of using CausalTime data differ from those we would draw using synthetic data.  If I were trying to figure out which method performs best using synthetic data, but the method I chose would actually perform worse on realistic data, that would be a very convincing argument for the value of CausalTime.  It also appears as though a citation is missing ("Besides, compared with the reported results from previous work (), ..."), so I'm unable to assess how the results actually do compare.

3.4 describes its purpose as describing how "to acquire the Actual Causal Graphs with high data fidelity".  However, the following paragraphs, up into the equation for the ACG in Equation 8, concerns generating the time series X, not the ACG. (the ACG is defined based on H, I_N, and J_N, none of which are defined in 3.4 prior to Equation 8) By the time you get to Equation 6, don't you already have the ACG? (since it relies on H) So it seems like the first sentence should instead read something like "To generate data from the Actual Causal Graph (ACG) with high data fidelity", rather than "To acquire the Actual Causal Graph (ACG) with high data fidelity"

This is minor, but there are some grammatical issues - for example, the first sentence of the 3rd paragraph of the introduction is a fragment.

**Questions:**

I don't see this mentioned anywhere - will code be provided upon publication?

In Definition 1, what is Y?  Is it X at time t? (What is the output of the neural network? I'm not seeing Y referenced after this section)

What was the motivation for using |AUC - 0.5|, rather than just AUC?


Edit post author response: Updating score from 6 to 8.

---

> ### Author Response · Authors · 2023-11-15
> **Point-by-point Response to Reviewer 3QnF (Part 1/2)**
>
> Thanks for your appreciation and questions! All your comments are of great help to improving our manuscript. Below are the point-to-point responses.
>
> > Q1: My biggest confusion/concern about this work is the inclusion of the residual term. Clearly, from Table 2, the residual term plays a massive role in producing data that looks like the original data. However, it seems as though the presence of the residual term simply means that every time series depends on every other time series. While it makes sense that including more variables would allow for more accurate model fitting, it's not clear to me that the resulting data actually reflects the causal dynamics of the supposed ground truth graph. Looking at Equations 6 and 7, it doesn't look like the residual term is down-weighted or anything to reduce its contribution. Substituting Equation 7 into Equation 6, it actually looks as though the primary f_{theta_i} terms cancel, leaving us with the first term in Equation 7 plus the noise term, which essentially means that we're just ignoring which variables are actually parents of x_i and just including everything. Am I misunderstanding or misinterpreting something? It looks to me like a ground truth graph is generated, and realistic-looking data is generated, but the realistic-looking data doesn't actually come from that graph. This is the main reason my score is a 6 rather than an 8, and if this is cleared up satisfactorally, I'd be happy to raise my score.
>
> A1: Thank you for your appreciation of our work and valuable questions. We are sorry for the confusion in Sec. 3.4. Although we split the NAR model into causal terms, residual terms, and noise terms to get a sparse causal graph, the time-series are generated from their combination, i.e., the original NAR model.  Specifically, we achieve this in three steps, which are also visualized in an additional Fig. 2:
>
> 1. Identify causal terms in Section 3.3, reflecting the **"major part"** of the causal relationship.
> 2. Split the NAR model into causal terms, residual terms, and noise terms, with the residual term representing the **"remaining"** causal relationship and the noise term accounting for randomness in the data. Since combining Eqn. 5 and 6 we get the original NAR model, the fidelity is preserved.
> 3. Generate new time-series with the NAR model and **treat the residual terms $\hat{x}^{\text{R}}$ as independent time-series**, resulting in $2N$ instead of the original $N$time-series and a $2N\times 2N$ ground-truth graph (ACG), as defined in Eqn. 7.
>
> Therefore, the time series generated by NAR is realistic, and more importantly, does actually come from the ground-truth causal graph. In conclusion, the intention for introducing independent residual terms is to ensure i) the original NAR model is unchanged and ii) the ground truth causal graph is not fully connected and represents the **"major parts"** of causal relationships. To help more readers understand, we have revised Section 3.4 by adding more intuitive explanations for Eqn. 5-7 and illustrative Fig. 2.
>
> We are not sure if this clears up your confusion. If not, we would be happy to answer more questions!

---

> ### Author Response · Authors · 2023-11-15
> **Point-by-point Response to Reviewer 3QnF (Part 2/2)**
>
> > Q2: A bit more analysis for the final evaluation would be helpful. The authors point out that, when evaluated on synthetic data in prior work, the scores are higher across the board. While that's interesting, I'm much more interested in if the conclusions we would draw as a result of using CausalTime data differ from those we would draw using synthetic data. If I were trying to figure out which method performs best using synthetic data, but the method I chose would actually perform worse on realistic data, that would be a very convincing argument for the value of CausalTime. It also appears as though a citation is missing ("Besides, compared with the reported results from previous work (), ..."), so I'm unable to assess how the results actually do compare.
>
> A2: Thank you for the suggestion and the careful reading. We have fixed the missing citation and added [1][2]. In these papers, the TSCD algorithms achieve AUROC higher than 0.9, some close to 1. Moreover, we can observe in Tab. 3 and the results in [1][2] that
>
> 1. Constraint-based approaches such as PCMCI achieve high performance on synthetic datasets (especially linear cases) but perform worse on CausalTime. This is probably due to the difficulty of conducting conditional independence tests in nonlinear or realistic scenarios.
> 2. Granger-causality-based approaches such as CUTS+ perform the best because of their neural network-based architecture. However, these methods, such as CUTS, CUTS+, NGC, eSRU, have limitations since they rely on the assumptions of causal sufficiency and no instantaneous effects.
> 3. CCM-based approach LCCM performs surprisingly well although the performance is not that good in synthetic datasets [1]. This proves that learning Neural-ODE-based latent processes greatly helps the real application of Convergent Cross Mapping.
>
> We have included these discussions in Section 4.3.
>
> [1] Y. Cheng et al., “CUTS: Neural Causal Discovery from Irregular Time-Series Data,” ICLR 2023.
>
> [2] A. Tank, I. Covert, N. Foti, A. Shojaie, and E. B. Fox, “Neural granger causality,” TPAMI.
>
> > Q3: 3.4 describes its purpose as describing how "to acquire the Actual Causal Graphs with high data fidelity". However, the following paragraphs, up into the equation for the ACG in Equation 8, concerns generating the time series X, not the ACG. (the ACG is defined based on H, I_N, and J_N, none of which are defined in 3.4 prior to Equation 8) By the time you get to Equation 6, don't you already have the ACG? (since it relies on H) So it seems like the first sentence should instead read something like "To generate data from the Actual Causal Graph (ACG) with high data fidelity", rather than "To acquire the Actual Causal Graph (ACG) with high data fidelity"
>
> A3: Many thanks for the suggestion. Firstly, $\mathbf{I}_N$ is the Identity matrix and $\mathbf{J}_N$ is the all-one matrix, we have added the explanation after Eqn 8. Secondly, the notion of ACG relies on splitting the NAR model into causal terms, residual terms, and noise terms, which are explained in detail in Section 3.4. Hence, in Eqn. 6, we still don't have ACG.
>
> However, we also agree that the presentation of Section 3.4 can be improved and have included more explanations. Specifically, we made the following revisions:
>
> 1. Alter the subtitle of Section 3.4 to "Splitting the NAR Model to Acquire Actual Causal Graph and Realistic Time-series".
> 2. Add explanations of why introduce independent residual terms, along with additional illustrative Fig. 2.
> 3. Add a more intuitive explanation for Eqn. 5-7.
>
> > Q4: This is minor, but there are some grammatical issues - for example, the first sentence of the 3rd paragraph of the introduction is a fragment.
>
> A4: Thank you for the careful reading. We have fixed these issues and revised the manuscript thoroughly.
>
> > Q5: I don't see this mentioned anywhere - will code be provided upon publication?
>
> A5: The code for the CausalTime pipeline is already in the supplementary file and will be released upon publication. Moreover, we have created a website at www.causaltime.cc to help researchers use our benchmark.
>
> > Q6: In Definition 1, what is Y? Is it X at time t? (What is the output of the neural network? I'm not seeing Y referenced after this section)
>
> A6: Sorry for the confusion. In Definition 1, $\mathbf{y}$ is only used to denote the output or prediction of the neural network. To avoid confusion, we have removed the notation $\mathbf{y}$ here.
>
> > Q7: What was the motivation for using |AUC - 0.5|, rather than just AUC?
>
> A7: Thank you for the question. We use discriminative scores to examine if a neural network classifier can distinguish generated time-series from the real ones. We can conclude that the generated time-series is "real" if the AUROC score is close to 0.5 and that they are not "real" if AUROC $\rightarrow$ 1. As a result, we use |AUC - 0.5| for better illustration.
>
> Thanks again for your questions. Please let us know if you have any more questions!

---

> > ### Comment · Reviewer_3QnF · 2023-11-22
> >
> > Thanks for your detailed response!  (apologies for the formatting of my 'Weaknesses' section - I didn't notice that it all got combined into a giant paragraph!)
> >
> > The clarifications and modifications made to the paper go a long way towards clearing up my confusion and concerns.  I'm happy to raise my score in response.

---

> > > ### Author Response · Authors · 2023-11-22
> > > **Thanks!**
> > >
> > > Thank you so much for your appreciation of this work!

---

### Official Review · Reviewer_jMsi · 2023-11-03

**Soundness:** 2 fair
**Presentation:** 4 excellent
**Contribution:** 3 good
**Rating:** 8
**Confidence:** 4

**Summary:**

The paper introduces a time series simulator that utilizes neural networks to closely match a real dataset. This simulator allows its users to manipulate the causal graph of the process by altering the neural network pathways, or by setting less significant inputs to zero. An individual neural network is employed for each generated time series, thereby encoding a stationary "family" within the causal graph of the Dynamic Bayesian Model-like structure. The simulator's construction and justification draw from various existing methodologies. The paper demonstrates that the datasets generated in this manner do indeed share the traits of the original time series, especially when evaluated through their nonlinear embeddings such as t-SNE. Furthermore, the paper benchmarks nine recent deep learning-based models against each other, using data produced by the simulator trained on three different real datasets.

*I raised my score after rebuttal*

**Strengths:**

1. The paper is well-written, with concepts and methodologies clearly explained, making it accessible and comprehensible.
2. It addresses a significant need in the field of causal time series modeling, offering a solution to a complex problem.
3. The evaluation of the proposed approach is relatively comprehensive, even though it is somewhat one-sided (as addressed in the paper's weaknesses). It includes benchmarking against several competing methods, which adds a comparative dimension to the analysis. Furthermore, an earnest attempt is made to measure the goodness of fit of the data generated by the proposed method, adding a quantitative validation to the study.

**Weaknesses:**

1. The only sanity check of performance of the simulator is the quality of the fit but not demonstration of the learned graphs and whether they are reasonable. Without an investigation into the graphs learned by the simulator we have a situation of severe under-determination of the system. What if, there is enough information in any reasonably sized subset of the prior time series to autoregressively fit any signal. What kind of graph the methods tested on the proposed simulated time series supposed to reconstruct? Table 3 does show non-random performance, so there's truth to it, but are the ACG graphs sparse, dense, high or low Markov order and do they even make sense from the "organic" perspective of the domain they have been generated for? These questions are left unanswered.
2. The focus on neural models in simulation and estimation is limiting. It would be best to show how other models are benchmark.

**Questions:**

1. Could you please clarify the notation for the ACG? Your A is represented as a $2N\times 2N$ matrix. The $2N\times N$ portion seemingly represents the "causal" and residual term mixing in the previous time step. However, the actual causal graph likely has an adjacency matrix of a $2N \tau_{max}\times N$ dimension to model edges from parents up to a lag of $\tau_{max}$. Does this notation imply that in this paper you were only considering Markov order 1 models?

2. I would appreciate if you could use more classical approaches, in addition to purely neural models, for benchmarking time-series causation. I suspect that the experiments may favor neural models over others (as noted in the weaknesses). It would be beneficial to include comparisons with the SVAR model, Granger Causality, the PC algorithm modified to work with time series, and similar models.

3. Could you please provide plots of the graphs that your simulator generates for each of the three real data test-case datasets?

4. Could you characterize the variability of your ground truth ACGs as a function of training your simulator model starting with different seeds?

5. I would appreciate an explanation of how the ground truth causal graphs in Figure 2 are combined.

---

> ### Author Response · Authors · 2023-11-15
> **Point-by-point Response to Reviewer jMsi (Part 1/2)**
>
> Thank you so much for the appreciation of our work and helpful comments! Below are the point-to-point responses.
>
> > Q1: The only sanity check of performance of the simulator is the quality of the fit but not demonstration of the learned graphs and whether they are reasonable. Without an investigation into the graphs learned by the simulator we have a situation of severe under-determination of the system. What if, there is enough information in any reasonably sized subset of the prior time series to autoregressively fit any signal. What kind of graph the methods tested on the proposed simulated time series supposed to reconstruct? Table 3 does show non-random performance, so there's truth to it, but are the ACG graphs sparse, dense, high or low Markov order and do they even make sense from the "organic" perspective of the domain they have been generated for? These questions are left unanswered.
>
> A1: Thank you for the valuable comments. The intention of this work is to generate time-series (i) with a unique ground truth causal graph for benchmarking and (ii) highly resembles the distributions of real data for evaluating the algorithm performance on real data. Hence, we might have various graphs to autoregressively fit the **original** time-series, **but for the generated version the latent graph is unique**, since we generate these time-series with the ACG. As for the rationality of the learned causal graphs, ​
>
> 1. In AQI and Traffic datasets, causal relationships are highly relevant to geometry distances, since nearby stations have mutual influences, so the extracted HCG and the subsequent ACG are directly from distance graphs, which align with common sense and are widely used [1][2].
> 2. In medical datasets, the HCG is extracted with DeepSHAP since it is hard to build reliable graphs from only prior knowledge. Although Shapley values might not match with causality, Shapley-value-based approaches are widely used in the field of medicine and are shown to capture features with actual important relationships [3][4].
>
> Consequently, the extracted graphs are built on existing extensive studies and are expected to be **close to those found in nature**. The generated ACG can be either sparse or dense, and the Markov order of different models varies, which are set as $\tau_{\text{max}}$ to fit the real time-series best and depend on the complexity of the underlying relationship. From the results, especially for AQI and Traffic, the resulting ACGs make sense from the "organic" perspective of the domain.
>
> [1] "Filling the G_ap_s: multivariate time series imputation by graph neural networks," ICLR 2022.
>
> [2] "Graph WaveNet for Deep Spatial-Temporal Graph Modeling." arXiv, May 31, 2019.
>
> [3] S. L. Hyland et al., "Early prediction of circulatory failure in the intensive care unit using machine learning," Nature Medicine.
>
> [4] H.-C. Thorsen-Meyer et al., "Dynamic and explainable machine learning prediction of mortality in patients in the intensive care unit: a retrospective study of high-frequency data in electronic patient records," The Lancet Digital Health.
>
> > Q2: The focus on neural models in simulation and estimation is limiting. It would be best to show how other models are benchmarks.
>
> > Q3: I would appreciate if you could use more classical approaches, in addition to purely neural models, for benchmarking time-series causation. I suspect that the experiments may favor neural models over others (as noted in the weaknesses). It would be beneficial to include comparisons with the SVAR model, Granger Causality, the PC algorithm modified to work with time series, and similar models.
>
> A2 / A3: Many thanks for the valuable suggestion. In addition to the PCMCI (which is a PC algorithm modified to work with time series) algorithm in the original manuscript, we will include more non-neural-network-based algorithms, including SVAR, Granger Causality, and NTS-NOTEARS. We have added the results to Section 4.3.

---

> ### Author Response · Authors · 2023-11-15
> **Point-by-point Response to Reviewer jMsi (Part 2/2)**
>
> > Q4: Could you please clarify the notation for the ACG? Your A is represented as a $2N\times 2N$ matrix. The $2N\times N$ portion seemingly represents the "causal" and residual term mixing in the previous time step. However, the actual causal graph likely has an adjacency matrix of a $2N \tau_{max}\times N$ dimension to model edges from parents up to a lag of $\tau_{max}$. Does this notation imply that in this paper you were only considering Markov order 1 models?
>
> A4: Thank you for the question. Our ACG $A$ ($2N\times 2N$) is actually the **causal summary graph**, which represents causal relations between time series without referring to time lags, and is common in TSCD [1], i.e., the causal effects may exist on multiple time lags $\tau, 0<\tau\leq \tau_{\text{max}}$. As a result, we are not only considering Markov order 1 models.
>
> [1] J. Runge, A. Gerhardus, G. Varando, V. Eyring, and G. Camps-Valls, “Causal inference for time series,” Nature Reviews Earth & Environment.
>
> > Q5: Could you please provide plots of the graphs that your simulator generates for each of the three real data test-case datasets?
>
> A5: Thank you. Visualization of three graphs is shown in Fig. 3, and we will plot the adjacency matrices in Supp. Fig. 8 for better demonstration.
>
> > Q6: Could you characterize the variability of your ground truth ACGs as a function of training your simulator model starting with different seeds?
>
> A6: Thanks for the helpful comments! We added experiments with 5 different seeds and measured the variability of ACGs with average standard deviation, shown in Supp. Section A.3.4. We can observe that the variability is relatively small, i.e., 5 graphs are similar.
>
> > Q7: I would appreciate an explanation of how the ground truth causal graphs in Figure 2 are combined.
>
> A7: Fig. 2 (Fig. 3 in the revised manuscript) is the visualization of causal graphs extracted from AQI, Traffic, and Medical. For AQI and Traffic, we overlay the ground truth causal graphs onto the map, since each node represents certain locations, and this figure shows that relationships between two variables are highly relevant to their geometric distances. However, for the Medical dataset, we simply plot the causal graph with the networkx package in python.
>
> We are not sure if this explains your question. If not, we would be happy to answer more questions!

---

### Official Review · Reviewer_7uMr · 2023-11-03

**Soundness:** 3 good
**Presentation:** 2 fair
**Contribution:** 3 good
**Rating:** 5
**Confidence:** 4

**Summary:**

This paper introduces a pipeline to generate time series with know causal graphs. The models considered are nonlinear auto-regressive with instantaneous causal effects. The procedure taken is as follows
 - run a causally disentangled neural network, here based on LSTM, on a real dataset
 - fit normalising flows to the residuals
 - sparsify the node-to-node relationships, the resulting graph $H$ is the HCG. This can be done using Shapley values or using prior domain knowledge.
 - to maintain higher data fidelity, the generated time series have double the dimension of the original data. Each time step consists of $\mathbf{x}$ and $\mathbf{x}^\text{R}$, where $\mathbf{x}^\text{R}$ represents the residual when predictions are made using the sparsified model with $H$. The advantage is that it is now possible to generate a new dataset that resembles the old as closely as possible and with a know, sparse causal graph.

Experimentally, the authors run their algorithm on 3 datasets and generate new causal benchmark time series. These are evaluated for fidelity with the original. Existing methods struggle on the resulting datasets.

**Strengths:**

- this paper presents a neat trick to generate a time series with a know causal graph that matches real data in distribution without knowing anything about the causal structure of the underlying data. This is achieved by doubling the dimension of the time series and having a "residual" stream as well as a real-variable stream
- the generated datasets present problems for existing temporal causal discovery methods

**Weaknesses:**

- this paper is only "causal" in a very weak sense. Indeed, given the "No instantaneous effects" assumption, it is possible to write down a dense causal graph in which all prior variables are parents of $\mathbf{x}_t$. Arguably, there is no true causality here, only the problem of learning a time series with *sparse* relationships, because the DAG constraint is automatically satisfied when instantaneous effects in the original time series are discounted.
- the primary novelty is the $N \rightarrow 2N$ trick to make a time series that fits the stated requirements. Whilst I think this is quite smart, the remainder of the paper is a concatenation of existing methods.
- this key idea is not explained particularly well in Sec 3.4.
- it is unknown how realistic the causal relationships used in CausalTime are. Indeed, the authors make no claim of doing causal discovery. Hence, the causal relationships used in CausalTime may be very different to those found in nature, implying that CausalTime datasets are not a good surrogate for causal discovery on real time series. Indeed, when using Shapley values, a correspondence between feature importance and causality is suggested which may be incorrect.

**Questions:**

- How does causal time fit in with methods that are designed to discover instantaneous relationships, like Rhino (Gong et al., 2022)? Could Rhino be applied to CausalTime datasets, and if so should it be included in the benchmarking? Based on your assumption of "No Instantaneous Effect", this would not be possible.
- The methods in the paper suggest that including the residual term is important. This, in turn, implies that "natural" causal graphs may be dense. E.g. all components of $\mathbf{x}_{t-1}$ affect $\mathbf{x}_t$. However, in CausalTime, the causal graph is forcibly sparsened to produce $H$, so as to present a more interesting problem to causal discovery algorithms. In Table 2, the inclusion of the residual term seems to reduce discriminative scores by a factor of at least 10, implying it is very necessary to get good reconstruction. Thus- is the forced sparsification used in CausalTime actually contrary to natural time series?

Gong, Wenbo, et al. "Rhino: Deep causal temporal relationship learning with history-dependent noise." arXiv preprint arXiv:2210.14706 (2022).

---

> ### Author Response · Authors · 2023-11-15
> **Point-by-point Response to Reviewer 7uMr (Part 1/2)**
>
> Thanks for your appreciation of this work and the careful reading! All your advice is of great help to us. Below are the point-to-point responses.
>
> > Q1: this paper is only "causal" in a very weak sense. Indeed, given the "No instantaneous effects" assumption, it is possible to write down a dense causal graph in which all prior variables are parents of $\mathbf{x}_t$. Arguably, there is no true causality here, only the problem of learning a time series with sparse relationships, because the DAG constraint is automatically satisfied when instantaneous effects in the original time series are discounted.
>
> A1: Thank you for your insightful comment. We would like to clarify our contribution in the following two aspects:
>
> 1. **Why include "no instantaneous effects" assumption:** We intend to develop a realistic benchmark to test various TSCD algorithms, some of which support instantaneous effects while a variety do not (e.g. CUTS, NGC, PCMCI). As a result, we choose to include the most common assumptions, e.g., No instantaneous effects, Causal Sufficiency, Causal Stationarity. Including instantaneous effects in the generation process is indeed a good idea, but it's beyond the scope of this work, so we include this direction in future works in Section 6.
> 2. **Why dense causal graph:** Actually, in CausalTime a dense prediction model **does not imply dense causal graphs** (also see Q6). Moreover, "True Causality" is usually inaccessible and we alternatively present a solution to get around this difficulty by splitting the "remaining part" (which **may or may not** contain causal effects) into the residual term. Mathematically, the sparse ACG is indeed the ground truth causal graph for the generated time-series (not the original input time-series). We have added illustrative Fig. 2 to clarify the misunderstanding.
>
> > Q2: the primary novelty is the $N \rightarrow 2N$ trick to make a time series that fits the stated requirements. Whilst I think this is quite smart, the remainder of the paper is a concatenation of existing methods.
>
> A2: Thank you for your comments. Indeed, our pipeline uses existing tools, e.g., normalizing flow, DeepSHAP, to fit noise distributions or extract important features. However, we would like to clarify that our contribution lies in the overall architecture and the fact that CausalTime is the first TSCD benchmark that is 1) realistic, 2) with ground truth graphs, and 3) generalizable to diverse fields.
>
> > Q3: this key idea is not explained particularly well in Sec 3.4.
>
> A3: Thank you. We have updated Section 3.4 to include more explanation. Specifically, we made the following revisions:
>
> 1. Alter the subtitle of Section 3.4 to "Splitting the NAR Model to Acquire Actual Causal Graph and Realistic Time-series".
> 2. Add explanations of why introduce independent residual terms, along with illustrative Fig. 2.
> 3. Add a more intuitive explanation for Eqn. 5-7.

---

> ### Author Response · Authors · 2023-11-15
> **Point-by-point Response to Reviewer 7uMr (Part 2/2)**
>
> > Q4: it is unknown how realistic the causal relationships used in CausalTime are. Indeed, the authors make no claim of doing causal discovery. Hence, the causal relationships used in CausalTime may be very different to those found in nature, implying that CausalTime datasets are not a good surrogate for causal discovery on real time series. Indeed, when using Shapley values, a correspondence between feature importance and causality is suggested which may be incorrect.
>
> A4: Thank you for the question. For many fields, including weather, traffic, or medicine, true causality is extremely hard to acquire (or maybe even impossible). If we try to discover causal relationships with another TSCD algorithm, we are faced with the dilemma of "using a TSCD algorithm to build a synthetic dataset to test TSCD algorithms". Therefore, we explicitly bypass doing causal discovery.
>
> As for the rationality of the used causal relationships, ​
>
> 1. In AQI and Traffic datasets, causal relationships are highly relevant to geometry distances, since nearby stations have mutual influences, so the extracted HCG and the subsequent ACG are directly from distance graphs, which align with common sense and are widely used [1][2].
>  2. In medical datasets, the HCG is extracted with DeepSHAP since it is hard to build reliable graphs from only prior knowledge. Although Shapley values might not exactly match with causality, Shapley-value-based approaches are widely used in the field of medicine and are shown to capture features with actual important relationships [3][4].
>
> Consequently, the extracted graphs in our three subsets are built on existing extensive studies and are expected to be **close to those found in nature**.
>
> [1] "Filling the G_ap_s: multivariate time series imputation by graph neural networks," ICLR 2022.
>
> [2] "Graph WaveNet for Deep Spatial-Temporal Graph Modeling." arXiv, May 31, 2019.
>
> [3] S. L. Hyland et al., "Early prediction of circulatory failure in the intensive care unit using machine learning," Nature Medicine.
>
> [4] H.-C. Thorsen-Meyer et al., "Dynamic and explainable machine learning prediction of mortality in patients in the intensive care unit: a retrospective study of high-frequency data in electronic patient records," The Lancet Digital Health.
>
>
> > Q5: How does causal time fit in with methods that are designed to discover instantaneous relationships, like Rhino (Gong et al., 2022)? Could Rhino be applied to CausalTime datasets, and if so should it be included in the benchmarking? Based on your assumption of "No Instantaneous Effect", this would not be possible.
>
> A5: Thank you for the questions. Our CausalTime does assume "No Instantaneous Effect" in the generation process, i.e., no $x_{t,i}\rightarrow x_{t,j}, i\neq j$ in the ground truth graph. However, testing on methods like Rhino is still possible by only testing the **time lagging parts**. We have added the citation of Rhino in the related works section and have added comparison experiments on Rhino in Section 4.3. It is indeed a worthwhile and feasible extension to support the **testing of the instantaneous parts**, and we have included this in the limitation of Supp. Sec. A.1.1 and future work in Sec. 6.
>
> > Q6: The methods in the paper suggest that including the residual term is important. This, in turn, implies that "natural" causal graphs may be dense, E.g., all components of ${x}_{t-1}$ affect $x_t$. However, in CausalTime, the causal graph is forcibly sparsened to produce $H$, so as to present a more interesting problem to causal discovery algorithms. In Table 2, the inclusion of the residual term seems to reduce discriminative scores by a factor of at least 10, implying it is very necessary to get good reconstruction. Thus- is the forced sparsification used in CausalTime actually contrary to natural time series?
>
> A6: Thanks for the interesting question. Indeed, the residual term implies that all components of $x_{t-1}$ are included to predict $x_{t}$. However, **not every variable in the prediction model necessarily means causal effect**, e.g., zero-value weights in a linear prediction model. The residual terms in CausalTime represent causal parents that are not major parts that **may or may not** affect ${x}_{t}$, so the "dense" prediction models in our pipeline do not imply that "natural" causal graphs are dense.
>
> Anyway, this is a very important aspect of the CausalTime pipeline, and we deeply thank the reviewer for this insightful question. We have added these discussions to Sec. 3.4 and changed "fully connection graph" to "fully connection prediction model" to avoid confusion.
>
> Thanks again for your time and we would be happy to answer any additional queries you may have!

---

> > ### Comment · Reviewer_7uMr · 2023-11-23
> > **Reply to authors' response**
> >
> > Thank you very much for your reply.
> >
> > Your responses to Q1-5 were insightful but did not substantially alter my opinion. The "no instantaneous effects" assumption is simply a working assumption of the paper, which is ok.
> >
> > > However, not every variable in the prediction model necessarily means causal effect, e.g., zero-value weights in a linear prediction model
> >
> > This is true, however, in such a case the error that one obtains with and without the predictor would be the same. In your case however, the error substantially drops.

---

> ### Author Response · Authors · 2023-11-23
> **Reply**
>
> Thank you for your response.
>
> > Q: in such a case the error that one obtains with and without the predictor would be the same.
>
> A: The error obtained without these predictors is calculated by deleting **all these predictors**. When a part of them have causal effects and the others do not, the error can still substantially drops.
>
> Thanks again for your time and we would be happy to answer any additional questions!

---

### Author Response · Authors · 2023-11-15
**Official Response to All Reviewers**

We deeply thank the reviewers for their appreciation of this work and insightful feedback. These suggestions greatly help to improve our work. In the following discussions, we individually address the reviewers’ comments.

We would like to clarify that our contribution exists in the overall architecture and building the first TSCD benchmark that is 1) realistic, 2) with ground truth causal graphs, and 3) generalizable to diverse fields. We briefly list the manuscript revisions as follows. In the manuscript, the updates are highlighted in blue.

For writing, we cleared the misunderstanding and improved the descriptions from three aspects:
1. **Basic Ideas:** add intuitive explanations in Section 3.4 and A.1.3, add illustrative Fig. 2, update the caption of Fig. 1, and reduce some mathematical details to better show the basic idea and key steps.
2. **Limitations:** discuss the assumptions and limitations in Supp. Section A.1.1.
3. **Analysis:** add more analysis of the existing algorithms in the experiments section.

Moreover, we have performed three suggested experiments and **added them to the manuscript**. The experiments include:

1. Perform additional quantitative comparison to more existing algorithms, including Rhino, SVAR, Granger Causality, and NTS-NOTEARS.
2. Characterize the variability of the extracted graphs by setting different seeds.
3. Introduce missing data entries and hidden confounding to CausalTime, and test the TSCD algorithms when faced with such issues.

We hope our revised manuscript and additional experiments help to provide further clarity. Please let us know if any more improvements can be made!

---

### Author Response · Authors · 2023-11-21
**Discussion Stage About to End**

Dear Reviewers and Area Chair,

The end of the discussion phase is approaching. We deeply thank the reviewers for their insightful comments and took significant efforts to address the reviewers' concerns.

However, we are still waiting for the replies from the 1st, 2nd and 3rd reviewers. Could you please take a look at our responses and let us know if any more improvements can be made?

We sincerely thank you for your efforts in reviewing our paper. Your insightful and constructive comments really help us.

Thanks, Authors

---

### Meta-Review · Area_Chair_2xSt · 2023-12-05

**Metareview:**

This paper introduces an algorithm to time-series causal discovery benchmark datasets with know causal graphs. The idea is to first fit a neural auto-regressive model, then extract a sparse graph, and finally use the sparse graph to modify the auto-regressive model and to generate residuals, and the resulting "splitting" approach maintains data fidelity while returning an "actual" causal graph that can be used for benchmarking.

Reviewers all agree that the paper contributes significantly towards addressing the lack of benchmark datasets for time-series causal discovery, and commend on the neat trick of producing a usable "actual causal graph" without knowing the ground truth. Some concerns are raised regarding assumptions, e.g., the exclusion of instantaneous effects, as well as how the benchmark generation method can cover causal graphs that are often seem in real-world datasets.

I personally think it is brilliant to have more benchmarks and this paper provides a nice way of generating them. However, the concern of whether the constructed "actual causal graphs" cover typical real-world causal graphs is also important. Iterating research developments on not-so-representative benchmarks can lead to misconceptions on research progress. I hope this point will be discussed in the final camera ready of this paper.

**Justification For Why Not Higher Score:**

It's a pity that the provided benchmarks are not very large-scale like vision or NLP data. Also the concern of whether the constructed "actual causal graph" covers typical real-world causal graphs is also important.

**Justification For Why Not Lower Score:**

I do think the causal discovery field lacks enough number of benchmarks so this paper is definitely going on the right direction.

---

### Decision · Program_Chairs · 2024-01-16

Accept (poster)